# Mechanoregulated inhibition of formin facilitates contractile actomyosin ring assembly

Dennis Zimmermann[1], Kaitlin E. Homa[1], Glen M. Hocky[2], Luther W. Pollard[3], Enrique M. De La Cruz[4], Gregory A. Voth[2], Kathleen M. Trybus[3] & David R. Kovar[1,5]

Cytokinesis physically separates dividing cells by forming a contractile actomyosin ring. The fission yeast contractile ring has been proposed to assemble by Search-Capture-Pull-Release from cytokinesis precursor nodes that include the molecular motor type-II myosin Myo2 and the actin assembly factor formin Cdc12. By successfully reconstituting Search-Capture-Pull in vitro, we discovered that formin Cdc12 is a mechanosensor, whereby myosin pulling on formin-bound actin filaments inhibits Cdc12-mediated actin assembly. We mapped Cdc12 mechanoregulation to its formin homology 1 domain, which facilitates delivery of new actin subunits to the elongating actin filament. Quantitative modeling suggests that the pulling force of the myosin propagates through the actin filament, which behaves as an entropic spring, and thereby may stretch the disordered formin homology 1 domain and impede formin-mediated actin filament elongation. Finally, live cell imaging of mechano-insensitive formin mutant cells established that mechanoregulation of formin Cdc12 is required for efficient contractile ring assembly in vivo.

[1] Department of Molecular Genetics and Cell Biology, The University of Chicago, 920 E. 58th St., CSLC 212, Chicago, IL 60637, USA. [2] Department of Chemistry, The James Franck Institute and Institute for Biophysical Dynamics and Computation Institute, The University of Chicago, 5735 S. Ellis Ave., Searle Chemistry Laboratory 231, Chicago, IL 60637, USA. [3] Department of Molecular Physiology and Biophysics, University of Vermont, 149 Beaumont Ave., HSRF 130, Burlington, VT 05405, USA. [4] Department of Molecular Biophysics and Biochemistry, Yale University, PO Box 208114, 266 Whitney Ave., New Haven, CT 06520-8114, USA. [5] Department of Biochemistry and Molecular Biology, The University of Chicago, 920 E. 58th St., CSLC 212, Chicago, IL 60637, USA. Kaitlin E. Homa and Glen M. Hocky contributed equally to this work. Correspondence and requests for materials should be addressed to G.A.V. (email: gavoth@uchicago.edu) or to K.M.T. (email: Kathleen.trybus@med.uvm.edu) or to D.R.K. (email: dkovar@bsd.uchicago.edu)

Key insights into the mechanism of contractile ring assembly have been obtained from seminal studies using the unicellular fission yeast *Schizosaccharomyces pombe* (reviewed in refs. [1, 2]). Identification of the involved proteins, the order of their recruitment and knowledge of their individual biochemical properties allowed the proposal of the quantitative Search-Capture-Pull-Release model[3], which provided insight into how membrane-bound ring precursor protein assemblies (cytokinesis nodes) mediate the formation of the contractile actomyosin ring. However, it remains elusive how the ring precursor proteins function in combination and regulate each other at the molecular level. Two of the approximately seven cytokinesis node components[4], the actin assembly factor formin Cdc12 and the type-II myosin motor Myo2, are thought to be necessary and sufficient to facilitate ring assembly via node coalescence[5–7]. The Search-Capture-Pull-Release model[3] posits that the node-bound actin assembly factor formin Cdc12 nucleates and elongates 'searching' actin filaments using the cytoplasmic pool of profilin-actin, while remaining continuously associated with the elongating actin filament barbed end. The type-II myosin motor Myo2-associated with a neighboring node then 'captures' the 'searching' filament and 'pulls' on the filament, thereby bringing neighboring nodes closer together before their attachment is 'released' by filament severing[3]. However, this behavior has never been recapitulated in vitro, leaving unknown the underlying molecular mechanisms and ensemble properties of components facilitating node condensation. In this study, we demonstrate the first minimal component reconstitution of the Search-Capture-Pull model for contractile ring assembly. We discovered that the application of sub-piconewton forces by the physiological force generator myosin Myo2 to formin Cdc12-bound actin filaments results in the reversible mechano-inhibition of Cdc12's ability to processively elongate actin filaments. Mechanistically, we identified the formin homology 1

(FH1) domain of Cdc12 as the region relaying the force-sensitive response. Quantitative modeling suggests that the applied tensile force propagates through the actin filament, which behaves as an entropic spring, and thereby may stretch the disordered FH1 domain and impede formin-mediated actin filament elongation over relatively large distances. Finally, live cell imaging of mechano-insensitive formin mutant cells established that mechano-inhibition of formin Cdc12 is required to effectively condense contractile ring precursors, thereby enabling efficient cytokinesis in vivo. These results open up a new area of investigation linking cytokinesis directly to cytoskeletal mechanotransduction, a phenomenon that may play a pivotal role in the regulation of other important cell biological processes that necessitate contractile actomyosin networks (e.g., cellular apical constriction during tissue morphogenesis or embryonic germ-band extension)[8–11].

## Results

**The contractile ring formin Cdc12 acts as mechanosensor.** We used total internal reflection fluorescence microscopy (TIRFM) to follow in vitro-reconstituted Search-Capture-Pull reactions containing 1 μm diameter biospheres (beads: node biomimetics) functionalized with either purified formin homology 1 (FH1) domain-anchored ring formin Cdc12(FH1FH2) or ring myosin Myo2 (Fig. 1a). We found that Cdc12- and Myo2-associated beads are sufficient to reconstitute Search-Capture-Pull in vitro. In a representative Search-Capture-Pull event (Fig. 1b, Supplementary Fig. 1a and Supplementary Movies 1 and 2), an actin filament processively elongating (Search) from a bead-associated Cdc12 encounters a Myo2-associated bead, and the successful engagement (Capture) of the Myo2 bead results in the coalescence of both beads. The average speed at which beads coalesce ($93 \pm 17$ nm s$^{-1}$ Mean $\pm$ s.e.m., $n = 11$ independent experiments,

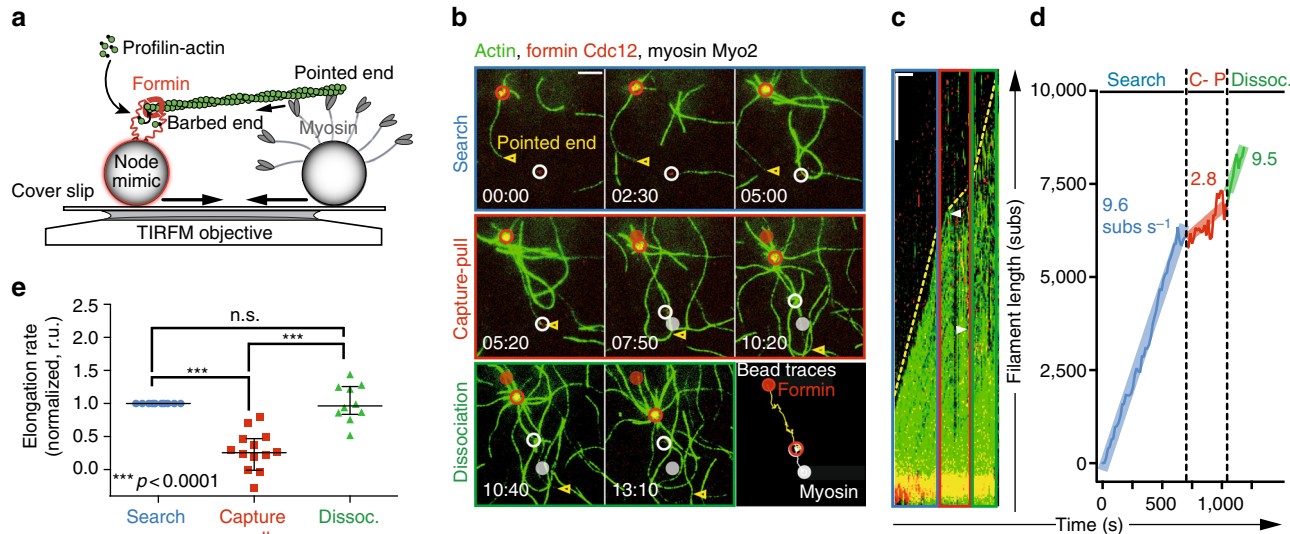

**Fig. 1** Formin Cdc12-mediated actin assembly is inhibited by myosin Capture-Pull. **a** Schematic of reconstituted Search-Capture-Pull using node mimics, 1 μm microspheres coated with either formin Cdc12 or myosin Myo2. **b–d** Dual-color TIRFM micrograph time-series (min:s) (**b**, scale bar = 5 μm), Kymograph (**c**, scale bars = 200 s and 5 μm in x- and y-direction, respectively) and filament elongation rates **d**, of an actin filament elongating (searching, *blue*) from a Cdc12-associated bead (*red circle*) that is captured and pulled (*red*) by a Myo2-associated bead (*white circle*) before dissociating (*green*). For the time series, initial bead positions are marked with opaque filled circles. For the kymograph, *dotted yellow lines* mark filament elongation rates and *white arrowheads* mark the Myo2 bead position at the beginning and end of Capture-Pull (Supplementary Movie 1). *Shaded lines* represent the regression line fits for the respective filament elongation rate trace during the Search, Capture-Pull and Dissociation phase. **e** Normalized Cdc12-mediated actin filament elongation rates during Search-Capture-Pull and Dissociation ($n = 11$ independent Search-Capture-Pull events). One-way ANOVA with the Tukey's multiple comparisons test was performed on the entire data set (n.s., not significant when $p \geq 0.05$, ***highly significant with $p < 0.0001$). Average formin elongation rates are listed in Supplementary Table 1

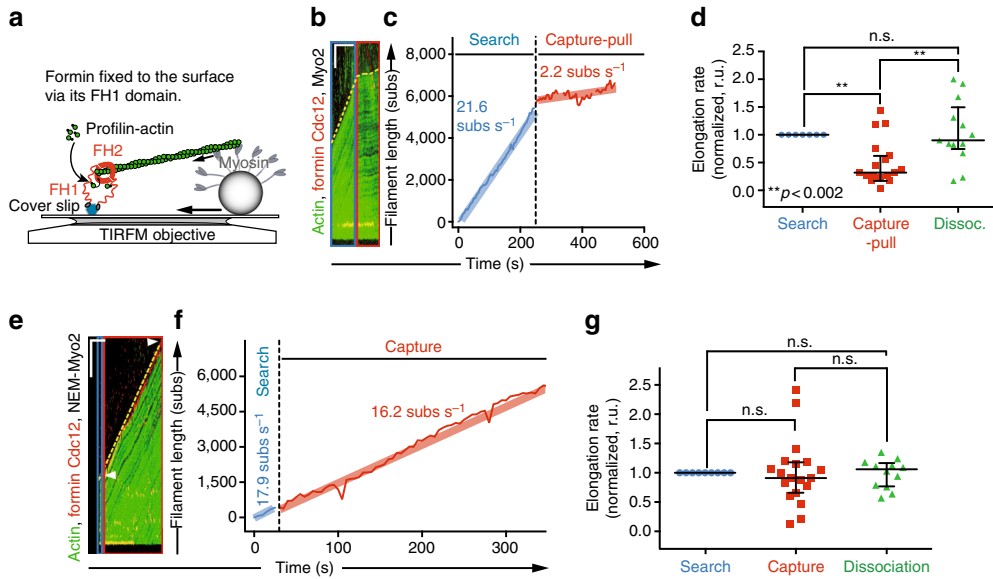

**Fig. 2** Active myosin Myo2 pulling is required to inhibit formin Cdc12. **a** Schematic of reconstituted Search-Capture-Pull with individual formin dimers fixed to the glass surface via the formin homology FH1 domain. **b–g** TIRFM kymographs **b**, **e**, corresponding elongation rates **c**, **f**, and normalized elongation rates (to pre-capture elongation rates) before (*blue*), during (*red*) and after (*green*) Search-Capture events **d**, **g**. Events are either surface-adhered Cdc12 and Myo2-associated beads **b–d** ($n = 7$ independent Search-Capture-Pull events), or surface-adhered Cdc12 and NEM-modified Myo2-associated beads **e–g** ($n = 11$ independent Search-Capture-Pull events) (Supplementary Movie 3). The Capture-Pull rate of 2.2 subs s$^{-1}$ in **c** derives from interpolating the regression across three Capture-Pull intervals that are separated by very brief (~ 10–15 s) myosin-bead dissociation events. For kymographs, *dotted yellow lines* mark filament elongation rates and *white arrowheads* mark the Myo2 bead position at the beginning and end of Capture-Pull. *Scale bars* = 200 s **b**, **e** and 5 μm **b** or 10 μm **e** in *x*- and *y*-direction, respectively. *Shaded lines* **c**, **f** represent the regression line fits for the respective filament elongation rate trace during the Search, Capture-Pull or Capture-only phase. One-way ANOVA with the Tukey's multiple comparisons test was performed on the entire data set (**d–g**, n.s., not significant when $p \geq 0.05$, **highly significant with $p < 0.002$). Average formin elongation rates are listed in Supplementary Table 1

Supplementary Table 2) is 6-fold slower than the reported velocity of Myo2 at saturated motor head densities ($542 \pm 11$ nm s$^{-1}$, $n = 10$, filament-gliding assay in Supplementary Fig. 2a)[7], recapitulating Myo2-driven in vivo node movement[3]. Considering geometrical constraints, quantitative immunoblotting estimates 9 to 12 bead-bound Myo2 heads are engaged with a filament at any given time (Supplementary Fig. 2b–d). This number is within a reasonable range of what is expected in vivo (seven to nine heads) (see Methods, Supplementary Fig. 2d)[12].

Although the biochemical properties of Cdc12 and Myo2 have been well established in isolation[7, 13–17], it is conceivable that their behaviors are altered in combination. During the Search stage, bead-bound Cdc12 elongates actin filaments at an average rate of $9.6 \pm 1.1$ subs s$^{-1}$ Median $\pm$ s.e.m. ($n = 11$ independent experiments, Fig. 1c, d and Supplementary Table 1), identical to previous reports for Cdc12 in solution or adhered to glass surfaces[14]. However, during Capture-Pull the average elongation rate is reduced ~ 3.5-fold to $2.8 \pm 0.9$ subs s$^{-1}$, representing a significant inhibition ($p < 0.0001$, One-way ANOVA with Tukey's multiple comparisons test) of Cdc12 activity (Fig. 1c–e, Supplementary Fig. 1c and Supplementary Table 1). Following dissociation of the Myo2-bead, the average elongation rate reverts back to $10.3 \pm 1.5$ subs s$^{-1}$ ($n = 11$, Fig. 1c–e and Supplementary Table 1).

**Mechano-inhibition of Cdc12 requires active pulling force**. To test Cdc12 mechanosensation at the single formin-dimer level, individual Cdc12(FH1FH2) dimers were directly anchored to the coverslip surface via the FH1 domain and subjected to Myo2-mediated pulling force (Fig. 2a). Identical to the inhibitory response of bead-anchored Cdc12, Myo2-mediated pulling of filaments elongating from single formin dimers anchored to

coverslips causes a similar 3.7-fold reduction in Cdc12-mediated F-actin elongation rate (Fig. 2a–d). Filaments elongate at $20.5 \pm 3.5$ and $5.6 \pm 1.8$ subs s$^{-1}$ before and during Capture-Pull, respectively ($n = 7$, Supplementary Table 1).

To test whether pulling force exerted by Myo2 is required to inhibit Cdc12, filaments elongating from Cdc12 molecules attached directly to the coverslip surface were captured by beads coated with chemically inactivated N-ethylmaleimide (NEM)-Myo2 (Supplementary Movie 3). NEM-Myo2 retains the ability to bind F-actin but is unable to exert a pulling force (Supplementary Fig. 10b, c)[18, 19]. NEM-Myo2 beads do not inhibit Cdc12-mediated actin filament elongation despite staying tightly bound to the filament (Fig. 2e–g, Supplementary Table 1 and Supplementary Movie 3). The dependence of Cdc12 inhibition on active pulling by Myo2 strongly suggests that Cdc12 acts as mechanosensor that responds negatively to pulling forces.

**Formin mDia2 remains fully active when pulled on by Myo2**. Prior work has shown processive elongation of actin filaments associated with mammalian formin mDia1 and budding yeast formin Bni1 is enhanced approximately 2-fold under hydrodynamic flow[20, 21]. We therefore tested whether Myo2-mediated pulling also inhibits the mammalian cytokinesis formin mDia2[22, 23]. Intriguingly, mDia2 (FH1FH2) is not inhibited, with similar elongation rates before ($28.2 \pm 3.8$ subs s$^{-1}$), during ($34.4 \pm 3.6$ subs s$^{-1}$) and after ($28.8 \pm 7.9$ subs s$^{-1}$) Capture-Pull ($n = 9$, Supplementary Table 1). We have also demonstrated that the difference is not due to mDia2's inherent faster actin filament elongation rate since mDia2 is also not inhibited at lower actin concentrations at which filament elongation rates are comparable to those of Cdc12 ($8.0 \pm 1.1$ subs s$^{-1}$, $n = 6$) (Fig. 3a–c, Supplementary Table 1 and Supplementary Movie 4).

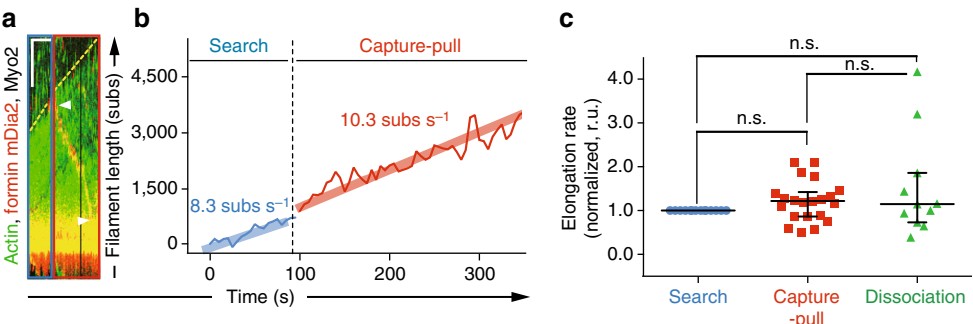

**Fig. 3** Formin mDia2 remains active during Capture-Pull. **a–c** TIRFM kymograph (**a**, scale bars = 100 s and 5 µm in x- and y-direction, respectively) and corresponding elongation rates (**b**), and normalized elongation rates (to pre-capture elongation rates) for multiple events before (*blue*), during (*red*) and after (*green*) Search-Capture-Pull **c** between surface-adhered formin mDia2 and Myo2-associated beads (n = 15 independent Search-Capture-Pull events) (Supplementary Movie 4). *Shaded lines* **b** represent the regression line fits for the respective filament elongation rate trace during the Search and Capture-Pull phase. One-way ANOVA with the Tukey's multiple comparisons test was performed on the entire data set (**c**, n.s., not significant when $p \geq 0.05$). For kymograph, *dotted yellow lines* mark filament elongation rates and white arrowheads mark the Myo2 bead position at the beginning and end of Capture-Pull. Average formin elongation rates are listed in Supplementary Table 1

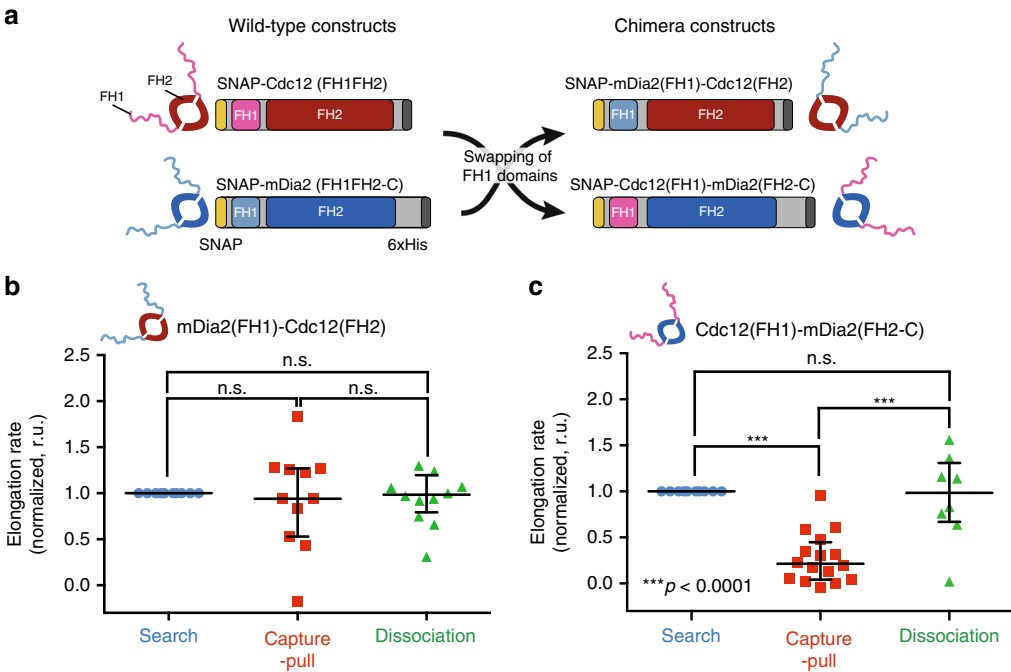

**Fig. 4** Mechanosensitive inhibition maps to formin Cdc12's FH1 domain. **a** Wild-type and chimeric formin constructs generated by exchanging the FH1 domains of Cdc12 and mDia2. **b, c** Normalized mDia2(FH1)-Cdc12(FH2)- **b** and Cdc12(FH1)-mDia2(FH2)-mediated **c** actin filament elongation rates before (*blue*), during (*red*), and after Capture-Pull (*green*) (n ≥ 10 independent Search-Capture-Pull events) of reconstituted Search-Capture-Pull between chimeric formin- and Myo2-associated beads. One-way ANOVA with the Tukey's multiple comparisons test was performed on the entire data set (**b, c**, n.s., not significant when $p \geq 0.05$, ***highly significant with $p < 0.0001$). Average formin elongation rates are listed in Supplementary Table 1

**Mechanosensitivity maps to Cdc12's FH 1 domain**. The formin homology 1 and 2 (FH1 and FH2) domains co-operate to processively elongate actin filaments[15]. The FH1 domains bind profilin-actin, which is subsequently transferred to the elongating barbed end that is associated with the torus-shaped FH2 domain dimer (Figs. 1a, 2a)[24]. It is conceivable that Myo2-dependent pulling forces could affect the unstructured FH1 domains[25], and/or Myo2-inflicted torsional forces might counter-act the apparent rotational behavior of the FH2 domains[26]. To determine which of the two domains senses and responds to mechanical stress, we tested the behavior during Capture-Pull of two chimera constructs containing exchanged FH1 domains of Cdc12 and mDia2 (Fig. 4a). Chimeric mDia2(FH1)-Cdc12(FH2) behaves in the same mechano-insensitive manner as wild-type

mDia2 (elongation rates before and during Capture-Pull: $11.4 \pm 2.2$ subs s$^{-1}$ and $9.5 \pm 2.0$ subs s$^{-1}$, n = 11) (Fig. 4b, Supplementary Fig. 3a, b and Supplementary Table 1). Conversely, just like wild-type Cdc12, Myo2-mediated pulling of chimeric Cdc12(FH1)-mDia2(FH2)-associated actin filaments inhibits their elongation significantly (~ 8-fold, $p < 0.0001$, One-way ANOVA with the Tukey's multiple comparisons test, elongation rates before and during Capture-Pull: $36.5 \pm 4.7$ subs s$^{-1}$ and $4.6 \pm 2.1$ subs s$^{-1}$, n = 10) (Fig. 4c, Supplementary Fig. 3c, d and Supplementary Table 1). Furthermore, neither wild-type Cdc12 (FH1FH2) or mDia2(FH1FH2) are inhibited by Myo2-mediated pulling forces when attached to the coverslip via their FH2 domains (Fig. 5a–c). Altogether these data suggest that the pulling force exerted by the Myo2 suffices to stretch key elements within

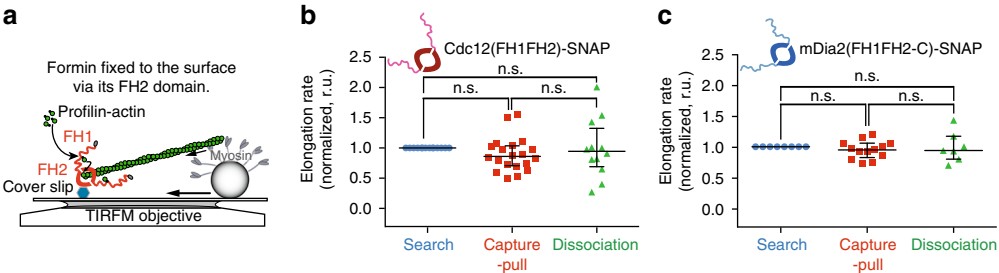

**Fig. 5** Formin Cdc12 remains active when its FH2 domain experiences pulling force. **a** Schematic of reconstituted Search-Capture-Pull with individual formin dimers fixed to the glass surface via the formin homology FH2 domain. **b, c** Normalized Cdc12(FH1FH2)-SNAP- **b** and mDia2(FH1FH2-C)-SNAP-mediated **c** actin filament elongation rates before (*blue*), during (*red*), and after Capture-Pull (*green*) ($n \geq 8$ independent Search-Capture-Pull events) of reconstituted Search-Capture-Pull between FH2-surface attached formin and Myo2-associated beads. One-way ANOVA with the Tukey's multiple comparisons test was performed on the entire data set (**b, c**, n.s., not significant when $p \geq 0.05$). Average formin elongation rates are listed in Supplementary Table 1

the unstructured Cdc12 FH1 domain away from the FH2 domain-associated barbed end, thereby preventing actin monomers from being added to the formin-bound barbed end of the filament.

**Cdc12's force response does not require filament tension**. To further explore the molecular mechanism(s) underlying Cdc12's mechanosensitive FH1 domain-dependent regulation, we adapted quantitative models of Vavylonis and co-workers[3, 26, 27] and designed coarse-grained simulations that mimic Search-Capture-Pull under in vitro and in vivo conditions (see Methods for details of the model, Supplementary Fig. 4 and Supplementary Movie 5). This allows us to capture the experimentally determined bead coalescence speeds, viscosity and bead drag (Supplementary Table 3), and to account for the observed actin filament and myosin motor behaviors and their ability to inhibit Cdc12-mediated elongation by active force. Importantly, in our reconstitution assays the beads (nodes) are substantially farther apart at the time of capture (2–25 μm) compared to nodes in vivo (~ 0.6 μm, Supplementary Fig. 4)[3]. The filaments are consequently bent, possibly affecting the force felt by the formin Cdc12. Intriguingly, our simulations reveal that the force transmitted from the myosin to the formin is relatively insensitive to apparent filament tension $T$ (filament contour length divided by direct node-to-node distance, Fig. 6a and Supplementary Fig. 5a, b). These findings further suggest that the formin load sensor must be sensitive enough to loads that are not adequate to straighten the formin-bound filament.

To probe this idea further, we categorized the in vitro experimental Capture-Pull events into low- and high-filament tension ($T$) regimes (Fig. 6b, c). For most Cdc12 Capture-Pull events the elongation rate is a third that of the myosin pulling velocity (~ 30 vs. ~ 100 nm s$^{-1}$, Supplementary Tables 1 and 2), suggesting that in those cases the myosin pulls its way toward the captured formin-bound barbed filament end at a faster rate than the filament is extending toward the myosin. Therefore, most of the Cdc12 Capture-Pull events result in rapid filament straightening (filled red symbols, Fig. 6b, c). We hypothesize that inhibition of Cdc12 ensures that filaments experience tension almost instantaneously, and that relatively low forces (~ 0.1 pN; Supplementary Fig. 5a) are sufficient to effectively inhibit Cdc12 without dissociating formin from the barbed end. Importantly, according to our mathematical model the forces propagated to filament-bound Cdc12 are largely independent of filament tension (Supplementary Fig. 5). This is also reflected by the observation that Cdc12 inhibition is effective over large distances (Supplementary Fig. 6a, b), and occurs even under conditions where the filament still has substantial slack (*open red symbols*,

Fig. 6b, c). Therefore, Cdc12 is the only formin identified so far that is inhibited by mechanoregulation, which requires relatively small (sub-piconewton) pulling forces.

**Cdc12 mechanoregulation facilitates ring assembly in vivo**. The quantitative Search-Capture-Pull-Release model predicted that Myo2-mediated inhibition of Cdc12 is required for proper ring assembly in fission yeast, without which cytokinesis nodes 'clump' and ring assembly is significantly delayed[3]. To experimentally test the physiological importance of Cdc12's mechano-inhibition, we engineered a 'mechano-insensitive' fission yeast strain by replacing the actin assembly FH1FH2 domains of Cdc12 with the FH1FH2 domains of mDia2 (Cdc12N-mDia2FH1FH2-Cdc12C: hereafter mDia2 or mDia2-GFP) (Supplementary Fig. 7a), which has similar actin assembly properties[15] but is not inhibited by Capture-Pull in vitro (Fig. 3c). As predicted, given that the N- and C-terminal regulatory regions are unchanged, mDia2-GFP properly co-localizes with Myo2's regulatory light chain Rlc1-tdTomato to contractile rings at the division site (Supplementary Fig. 7b). Formin mDia2 supports cell division with modest cytokinesis defects, reflected by an ~ 1.5-fold increase (from 22 to 32% for *cdc12* vs. *mDia2* cells) in the number of multi-nucleated cells, and a ~ 7-fold increase (from 4 to 28% for *cdc12* vs. *mDia2* cells) in cells that formed abnormal septa (Supplementary Fig. 7c–e).

As expected, mechanosensitive wild-type *cdc12* cells assemble nodes at the equatorial plane in the characteristic broad band (~ 3 μm wide) along the axis of the cell (Fig. 7a, b)[3]. Conversely, mechano-insensitive *mDia2* cells form nodes that subsequently collapse into clumps, yielding a significantly smaller band width of ~ 2 μm (Fig. 7a, b, and Supplementary Movies 6, 7). 3D surface plots revealed that the *mDia2* cell clumps contain multiple individual nodes with brighter fluorescence intensity peaks, while control *cdc12* cells display multiple individual fluorescence intensity peaks homogeneously dispersed at the equator (Fig. 7c). Further quantitative fluorescence image analysis revealed that *mDia2* cells contain more than twice as much F-actin material in the assembling ring than control *cdc12* cells (Fig. 7d, e). As discussed further below, we hypothesize that node clumping likely arises from entanglement of nodes through excessive F-actin generation by uninhibited formin (illustrated in Fig. 7f). Moreover, control *cdc12* cells initiate ring assembly ~ 2 min after spindle pole body separation and have formed fully mature rings after ~ 14 min, whereas *mDia2* cells take significantly longer (~ 21 min, $p < 0.00001$, two-sided student's $t$-test) to assemble mature rings (Fig. 7g, h).

Our mathematical modeling approach (see Methods, Supplementary Fig. 4 and Supplementary Movie 5) allowed us to

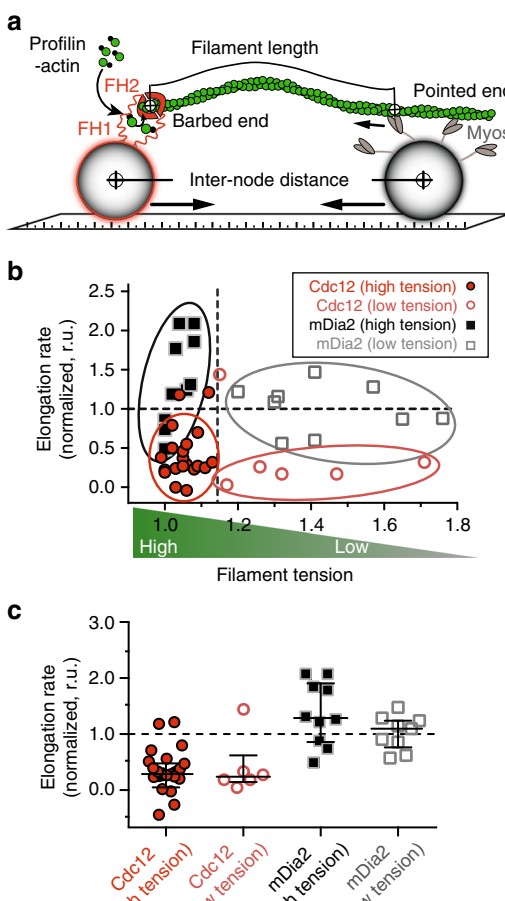

**Fig. 6** Formin Cdc12's force response does not require filament tension. **a** Illustration of filament tension $T$, the ratio of node-to-node contour filament length and formin node-to-myosin node distance. **b** Dependence of normalized formin elongation rates on filament tension. **c** Data from (b) summarized in a Box–Whisker plot. Elongation rates were normalized to the rate measured prior to myosin Capture-Pull. $n_{Cdc12} = 19$ independent Search-Capture-Pull events, $n_{mDia2} = 15$ independent Search-Capture-Pull events. Average formin elongation rates are listed in Supplementary Table 1

evaluate the impact of critical parameters such as differences in bead friction that can result in differences in drag forces in our in vitro experimental setup (Supplementary Figs. 5 and 8). Further, our mathematical model enabled us to study the potential role of formin mechano-inhibition for contractile ring assembly under in vivo-like conditions, where the formin-mediated filament elongation rate and drag coefficients are ~ 8- and ~ 100-times higher than in our in vitro setup, respectively. Without Cdc12 inhibition, even at saturated motor velocities of ~ 500 nm s$^{-1}$ (Supplementary Fig. 2a)[7], Myo2 assemblies are not sufficient to facilitate the rate at which nodes coalesce in vivo (30 nm s$^{-1}$)[3]. In this scenario, we estimate that Myo2 motor speeds >600 nm s$^{-1}$ would be required for node coalescence to overcome in vivo actin filament elongation rates (~ 75 subs s$^{-1}$)[3] (Fig. 7i). In striking contrast, given the mechanosensitive inhibition of Cdc12, the engaged Myo2 motors have to pull (under load) at only ~ 60 nm s$^{-1}$ in order to facilitate node coalescence at 30 nm s$^{-1}$ (Fig. 7i). With approximately seven to nine Myo2 heads available to engage at any given time in vivo (Supplementary Fig. 2d)[12], this represents a fairly realistic speed range and is in line with the previously proposed number of myosin heads required to obtain a quasi-processive pulling behavior (duty ratio 0.05)[7]. To conclude, mechano-inhibition of

node-anchored Cdc12 is likely required for proper contractile ring assembly during cytokinesis in dividing fission yeast cells (Fig. 8). In fission yeast cells containing the mechano-insensitive formin mDia2, ring precursor nodes collapse into clumps that impede and ultimately delay efficient contractile ring assembly significantly (Fig. 7h and Supplementary Movies 6, 7).

## Discussion

The actin cytoskeleton is capable of receiving, processing, transmitting and generating mechanical stresses, and has thus been long viewed as a central player facilitating diverse mechanotransduction pathways (recent reviews by refs. [28, 29]). We have recently begun to appreciate and probe the role of mechanical stresses in modulating the interaction of key regulatory actin-binding proteins with actin filaments, thereby controlling the assembly (e.g., formin and Arp2/3 complex) and disassembly (e.g., ADF/Cofilin) of individual actin filaments and entire F-actin networks[20, 21, 30, 31]. It was recently revealed that under hydrodynamic flow, tensile forces in the range of 0.1–2.5 pN can accelerate formin Bni1- (budding yeast) and mDia1- (mammalian) mediated F-actin elongation rates ~ 2-fold[20, 21].

In this study, for the first time we have reconstituted from purified components the previously proposed Search-Capture-Pull model for contractile ring assembly in dividing fission yeast cells[3]. We have made the important discovery that mechanical stress applied to actin filaments by the physiological force generator myosin results in the three- to four-fold mechano-inhibition of the contractile ring F-actin assembly factor formin Cdc12 (Fig. 1 and Supplementary Fig. 1). Mechanistically, we map formin Cdc12's mechanosensitivity to its FH1 domain (Fig. 4 and Supplementary Fig. 3). Using mathematical modeling and live cell imaging, we demonstrate that mechanoregulation of node-anchored Cdc12 ensures proper contractile ring assembly in fission yeast cells (Figs. 7 and 8). Further, Cdc12 remains active when myosin pulling force is applied directly to Cdc12's FH2 instead of its FH1 domain (Fig. 5b). We therefore hypothesize that applied pulling force stretches Cdc12's FH1 domain away from the FH2 domain-bound actin filament barbed end, thereby impeding the continuous transfer of new FH1 domain-bound actin subunits onto the elongating actin filament (Fig. 8, *red box*). For formins mDia1 and Bni1, the response to flow-induced mechanosensitivity occurs in a non-linear fashion[20, 21], making it difficult to directly compare the mechanosensitivities of those formins to that of Cdc12. However, it should be mentioned that formin Cdc12 responds to force in a manner that is not only of the opposite sign but also seems to be much more sensitive to force than formins Bni1 and mDia1 (3- to 4-fold change at ~ 0.1–1 pN vs. ~ 2-fold change in activity over a 3 pN range)[20, 21]. This force sensitivity, which could be >60-times higher per pN may be reflective of the difference in mechanism between the FH1-mediated mechanosensitivity of formin Cdc12's and the presumed FH2-dependent mechanosensitivity of formins Bni1 and mDia1[20, 21].

Our results are commensurate with previous simulations, which showed that inhibition of formin-mediated actin poly-merization results in more efficient contractile ring assembly[3]. Furthermore, a recent computational model provides an initial insight into the theoretical behavior of FH1 domains under tensile force[25]. The high disorder of FH1 domains causes steric inhibition towards the binding of profilin-bound actin monomers to specific sites (proline-rich stretches) on FH1[24, 25]. Specifically, tensile force may overcome this occlusion-type problem by untangling FH1 and thereby increasing the probability of actin monomer binding to FH1 (i.e., increase in actin monomer

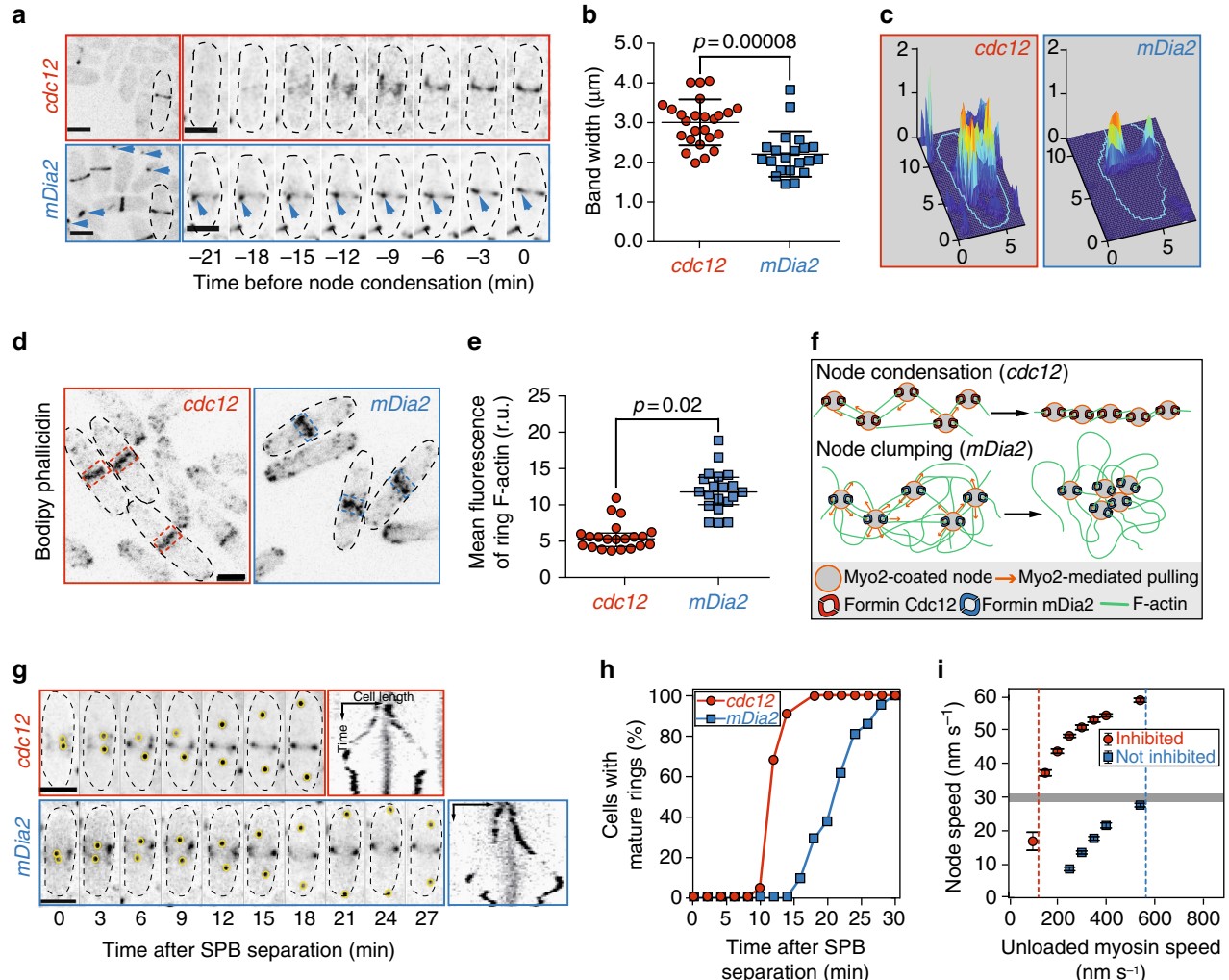

**Fig. 7** Myosin-mediated inhibition of formin Cdc12 facilitates contractile ring assembly in vivo. **a** Representative fields and time-series of dividing control (*cdc12*) and chimera mutant *mDia2* (Cdc12N-mDia2FH1FH2-Cdc12C) fission yeast cells expressing Rlc1-3GFP (Supplementary Movies 6 and 7). Cytokinesis node clumping is marked with blue arrowheads. *Scale bars* = 5 μm. **b** Average node band width ($3.0 \pm 0.6$ μm vs. $2.2 \pm 0.6$ μm for *cdc12* and *mDia2*, Mean ± S.D., $n \geq 21$ cells; $p = 0.00008$, and **c** 3D surface plots of representative node distributions, 8 min before ring maturation. **d** BODIPY-phallicidin staining of F-actin organization in *cdc12* (left) and *mDia2* (right) cells. Cell midzone used for quantification in **e** is boxed with dashed lines. *Scale bar* = 5 μm. **e** Plot of median F-actin fluorescence in the contractile ring for *cdc12* and *mDia2* cells. Median relative fluorescence, $5.6 \pm 0.5$ vs. $12.9 \pm 0.7$ for *cdc12* and *mDia2*, Median ± s.e.m., $n \geq 20$ cells; $p = 0.02$. One-sided (unpaired) Student's *t*-test was used to compare the two groups of data sets (**b**, **e**). **f** Cartoon model showing uninhibited actin assembly by mDia2 leads to node clumping by disorganized connections between multiple nodes. **g** Time-series (*scale bars* = 5 μm) and kymographs (*scale bars* = 5 μm and 5 min in x- and y-direction, respectively); and **h** efficiency ($n \geq 22$ cells) of contractile ring assembly in cells expressing spindle pole body (SPB)-marker Sad1-tdTomato (*yellow circles*) and Rlc1-tdTomato. **i** Simulations at in vivo conditions of the dependence of node coalescence speed (*y*-axis) on the Myo2 speeds required for driving node coalescence at the in vivo elongation rate of 75 subs s$^{-1}$. *Gray horizontal bar* indicates the reported in vivo speed of coalescing nodes (30 nm s$^{-1}$)[3]. Intercept of *dashed vertical red and blue lines* with the *x*-axis mark the myosin speed at which nodes would coalesce at the in vivo rate under conditions where formin undergoes inhibition (60 nm s$^{-1}$) or remains active (>600 nm s$^{-1}$), respectively. *Error bars* indicate s.d. of the mean deduced by jackknife resampling ($n = 24$–$48$ simulations at a given polymerization rate)

capture rate $k_{cap}$). Importantly, at the same time tensile force also pulls away profilin-binding sites from the FH2-bound barbed filament end, and thus decreases exponentially the delivery rate ($k_{del}$) at which FH1-bound actin monomers are added onto the filament end. Therefore, tensile force can alter profilin-actin's access to cryptic binding sites on FH1 domains, where, depending on its length and the relative position of the profilin-binding sites along the length of FH1, filament polymerization can either speed up or slow down. A comparative sequence analysis of Cdc12 and mDia2 FH1 domains (Supplementary Fig. 9) shows that in addition to a general lack of sequence similarity, Cdc12 FH1 is ~ 25% shorter and its two profilin-binding proline-rich tracks (PBT1 and PBT2) are considerably farther away from the

carboxy-terminal FH2 domain (distance to FH2 domain for PBT1 and PBT2 of Cdc12 vs. mDia2: 60 and 20 vs. 31 and 15 amino-acids). The computational model recently proposed by Bryant et al.[25] would predict that, under force, Cdc12 undergoes deceleration and mDia2 acceleration (Fig. 5 in ref. [25]), which is in good agreement with our experimental data (Figs. 1e, 2d and 3c).

Interestingly, while formin mDia2-mediated filament elongation rates are unchanged under low tension, 40% of the recorded events under high tension display a ~ 2.5-fold increase in elongation rate (filled black symbols in Fig. 6b, c). Similarly, processive elongation for formins mDia1 and Bni1 increases ~ 2-fold, albeit upon application of three- to five-fold higher force using hydrodynamic flow[20, 21]. In contrast to an FH1-dependent

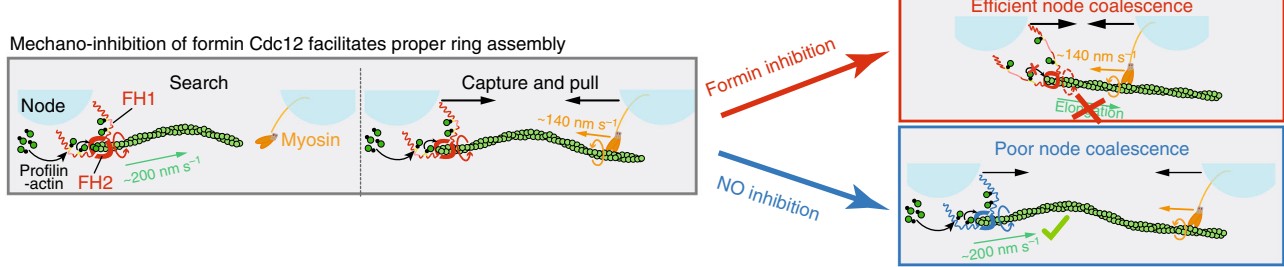

**Fig. 8** Cartoon model of how mechano-inhibition of formin Cdc12 is critical for facilitating proper ring assembly from cytokinesis nodes. By slowing down in vivo F-actin polymerization rates to a level where Myo2 pulling net velocity is sufficiently larger (Fig. 7i), mechano-inhibition of formin Cdc12 facilitates effective node coalescence (Fig. 7h). In addition, mechano-inhibition of formin Cdc12 prevents excessive F-actin generation, which would impede broad band formation once neighboring nodes are inter-linked upon capture (Fig. 7a–f)

mechanism mediating formin Cdc12's mechano-inhibition, the mechano-enhancement of formins mDia1 and Bni1 appears to directly depend on their FH2 domains, which have been hypothesized to be pulled open by increasing force resulting in accelerated addition of new actin subunits[20, 21]. Importantly, in our experiments where mDia2 was anchored to the surface via its FH2 domain (Fig. 5a), direct pulling by myosin on mDia2's FH2 domain has no significant impact on mDia2's baseline activity (Fig. 5c). This suggests that mDia2's potential mechano-enhancement may also depend on its FH1 domain, however much higher forces and/or filament tension may be required.

By analyzing the mean-square-displacement of myosin-coated beads prior to capture, we estimated the drag (i.e. friction) experienced by the beads in our in vitro reconstitution setup. Our analysis revealed that based on the diffusive bead behavior prior to capture (see Methods for details), events take place at a viscosity estimate of $\eta = 0.012$ Pa s, which is in good agreement with the theoretical dynamic viscosity of an aqueous solution containing 0.8% methylcellulose (400 cP) (Sigma #M0262 product information). However, in 75% of cases the beads showed sub-diffusive behavior, which we attribute to non-specific interactions of the beads with the coverslip. Given that these data are noisy and we do not have strong evidence for the precise mechanism underlying this sub-diffusion, we fit data from these sub-diffusing beads as if they are diffusing in order to extract an approximate average viscosity/drag coefficient experienced by the beads, although this procedure is not strictly correct. The effective viscosity extracted in this way is $\eta = 0.22$ Pa s, which is ~ 20-fold higher than our estimate of the viscosity of the medium. Using $F = v \times (6 \pi r \eta)$, we estimate a force of 0.1 pN ($F = 50$ nm s$^{-1} \times 6 \times 3.14 \times 500$ nm $\times 0.22 \times 10^{-6}$ pN s nm$^{-2}$) is required to bring two beads together at an average speed of 100 nm s$^{-1}$ (i.e., with the formin bead moving at 50 nm s$^{-1}$, Supplementary Table 1), which is in strong agreement with the results obtained from our computational model (Supplementary Fig. 8).

With relatively small forces ranging from 0.01 to 1 pN (Supplementary Fig. 5 and Supplementary Table 5), in most cases the applied force is far from Myo2's reported stall force (2 pN per myosin head)[3, 32], arguing that in our in vitro system the myosin beads operate under essentially load-free conditions where the myosin and formin beads are pulled together at the load-free myosin velocity of ~ 100 nm s$^{-1}$ (Supplementary Table 2). It has been shown that under load-free conditions, the myosin pulling speed depends strictly on the number of engaged heads[7]. At higher myosin densities, Myo2 velocities can saturate at speeds as high as 542 nm s$^{-1}$ (Supplementary Fig. 2a, compared with 535 nm s$^{-1}$ in Stark et al.[7]). Our biomimetic data indicate that ~ 10 myosin heads are available to engage with the captured

filament (Supplementary Fig. 2d), which is in strong agreement with the number that has been reported previously by Stark et al. (~ 10 heads required for speeds of 100 nm s$^{-1}$)[7].

Based on very recent quantitative high-speed fluorescence photoactivation microscopy (FPALM), ~ 10 Myo2 dimers (i.e., 20 myosin heads) are bound to one fission yeast cytokinesis node[12]. Taking into account geometrical constraints (for details, see Methods), this translates into seven to nine myosin heads proximal to the captured actin filament at a given time. Therefore, with 10 to 12 myosin heads available to engage a captured actin filament in our in vitro setup, the Myo2 node mimics used in this study compare well to cytokinesis nodes in vivo (Supplementary Fig. 2c, d). With significantly (100-times) higher drag forces on coalescing nodes in vivo (Supplementary Fig. 5c, d and 8) due to the ~ 25-fold higher cytoplasmic viscosity (~ 0.3 Pa s)[27, 33] and the additional (50-fold) drag that is created by the anchorage of nodes to the plasma membrane[34, 35], our mathematical modeling data provide additional support for the fact that during ring assembly cytokinesis nodes coalesce three to four times more slowly (~ 30 nm s$^{-1}$)[3] than observed in our experimental setup (~ 100 nm s$^{-1}$; Supplementary Table 2).

In addition to the myosin pulling force, compressive forces from the formin 'pushing' on the elongating actin filament could potentially factor into the net force that is applied on the formin and hence may affect its inhibition. Experiments where the diffusive behavior of catalytically inactive NEM-Myo2 beads (incapable of exerting active pulling force) during the capture phase was followed, convincingly show that filament-bound NEM-Myo2 beads exhibit a strictly (sub)diffusive behavior when compared to active NEM-Myo2 bead-mediated Capture-Pull (Supplementary Fig. 10b, c). We are confident that under the conditions tested, the force of formin-mediated actin polymerization is not sufficient to induce significant pushing drift on the myosin bead and concomitantly would be too small to contribute to formin Cdc12's mechano-inhibition under these conditions.

Interestingly, even with formin-to-myosin filament lengths reaching three-times (~ 30 μm, Supplementary Fig. 6b) the persistence length of F-actin, Cdc12 inhibition often results in instantaneous filament tension (filled red symbols, Fig. 6b, c). This could suggest that tension on the formin-bound filament might be a prerequisite for the mechano-inhibition of Cdc12. However, even under decreasing filament tension (i.e., increasing filament slack) conditions Cdc12 still undergoes significant inhibition (open red symbols, Fig. 6b, c), demonstrating that Cdc12 mechano-inhibition does not require filament tension. This directly emphasizes the notion that the relatively low forces (~ 0.1 pN) propagated to filament-bound Cdc12 are largely independent of filament tension, as is also shown using our

mathematical model (Supplementary Figs. 5a and 8a,b). We ascribe this phenomenon to the actin filament behaving as an entropic spring, whereby force from the myosin propagates through the filament via thermal fluctuations of the filament itself and regardless of how extended the filament is[36].

While filament tension is not required for mechano-inhibition of Cdc12, simulation and in vivo data re-establish that for two connected beads (in vitro) or nodes (in vivo) to efficiently coalesce, the connecting filament must be taught (Supplementary Movie 5, Fig. 7g–i). Therefore, for an actin filament to undergo straightening, the myosin has to pull its way toward the captured formin-bound barbed actin filament end at a faster rate than the actin filament is extending toward the myosin (Figs. 7i and 8). This is reflected by the in vitro finding where most Cdc12 (elongation rate is a third of the myosin speed) Capture-Pull events result in rapid filament straightening (filled red symbols, Fig. 6b, c). On the contrary, in vivo Cdc12 elongation rates are seven- to eight-fold faster than in vitro rates[3] and thereby likely two- to three-fold faster than myosin's pulling speed, creating a problem for dividing cells that can only be solved by slowing down (inhibiting) formin-mediated filament elongation upon myosin-mediated Capture-Pull (Fig. 8). Our mathematical modeling approach allowed us to further study the role of Cdc12's mechano-inhibition during contractile ring assembly under physiological conditions (i.e., 100-times higher drag forces and 8-times faster filament elongation rates)[3, 27, 33]. Importantly, our modeling data highlight the notion that mechano-inhibition of formin Cdc12 facilitates productive node coalescence by enabling myosin motor assemblies to efficiently pull together neighboring nodes in vivo (Supplementary Movie 5, Figs. 7i and 8).

Biochemical and super-resolution data suggest that the type-II myosin Myo2 in fission yeast associates with cytokinesis nodes as individual homo-dimers. Conversely, most non-muscle type-II myosins involved in applying the force on contractile rings, as well as other contractile networks (e.g., stress fibers), multimerize into bipolar filaments[37, 38]. While the number of heads per filament (∼ 28–58)[39] may be similar to the total number of Myo2 heads per cytokinesis node in fission yeast (∼ 20)[12], the total number of myosin motors that are at work during contractile ring assembly of multi-cellular eukaryotic cells remains largely unknown. This suggests that potential differences in myosin motor organization (mono- vs. bipolar, mono- vs. multimeric, clustered vs. dispersed) are likely to account for different stress magnitudes acting not only within the contractile ring but actomyosin-driven contractile processes in general (e.g., apical constriction and embryonic germ-band extension)[8–10], entailing different force responses by the different cytoskeletal mechanosensors (e.g., formin Cdc12 vs. mDia2). For example, members of the formin family are critically involved in assembling actin filaments for stress fibers and focal adhesions[40, 41], filopodia[42, 43], nuclear as well as perinuclear actin[44, 45], and the cell cortex[46, 47]. Hence, it is quite tempting to speculate that it is the employment of specific sets of actin assembly factors, which are specifically tailored to the underlying physical constraints of a particular cellular process, that endows cells to sense, process and respond to force. Therefore, it will be fascinating to elucidate the underlying mechanisms for mechanosensation of different sets of formins as well as other assembly factors (e.g., Ena/VASP family)[48, 49] that are involved in diverse force-generating cellular processes.

## Methods

**Buffers**. *Buffer A.* 10 mM imidazole, pH 7.0, 50 mM KCl, 1 mM MgCl$_2$, 1 mM EGTA, 50 mM DTT, 0.2 mM ATP, 15 mM glucose, 20 μg ml$^{-1}$ catalase, 100 μg ml$^{-1}$ glucose-oxidase, and 0.8% (wt/vol) methylcellulose (400 cP).

*Buffer B.* 0.5 mM MgCl$_2$ and 0.2 mM EGTA.

*Buffer C.* 2 mM Tris-Cl, pH 8.0, 0.2 mM ATP, 0.1 mM CaCl$_2$, 1 mM NaN$_3$ and 0.5 mM DTT.

*Buffer D.* 10 mM Hepes, pH 7.5, 100 mM KCl, 1 mM MgCl$_2$, 0.1 mM CaCl2 and 1 mM ATP.

*Buffer E.* 20 mM Hepes, pH 7.4, 200 mM KCl, 0.01% NaN$_3$, 1 mM DTT and 10% glycerol.

*Buffer F.* 300 mM NaCl, 10 mM imidazole pH 7.4, 5 mM MgCl$_2$, and 1 mM EGTA, 2 mM DTT, 7% w/v sucrose, 0.5 mM 4 -(2-Aminoethyl) benzenesulfonyl fluoride hydrochloride, 5 μg ml$^{-1}$ leupeptin, 0.5 mM phenylmethylsulfonyl fluoride and 0.4 mg ml$^{-1}$ benzamidine.

*Buffer G.* 300 mM NaCl, 10 mM imidazole pH 7.4, 1 mM EGTA, 1 mM NaN$_3$, 50% glycerol (vol/vol), 2 mM DTT and 1 μg ml$^{-1}$ leupeptin.

*Buffer H.* 50 mM Tris-Cl, pH 7.5 and 600 mM NaCl.

*Buffer I.* 25 mM imidazole, pH 7.4, 25 mM KCl, 4 mM MgCl$_2$, 1 mM EGTA and 10 mM DTT.

**Protein expression and purification**. Ca-ATP actin was purified from rabbit skeletal-muscle acetone powder (Peel Freez Biologicals, Rogers, AR)[50] and labeled on Cys374 with Oregon Green iodoacetamide (Life Technologies, Carlsbad, CA)[51], or on surface lysines with Alexa488-succinimidylester (Life Technologies)[52, 53]. Immediately before each experiment Ca-ATP actin was converted to Mg-ATP actin by adding 0.1 volumes of Buffer B.

Coding sequence for the Myo2 heavy chain (HC) was cloned into the *Sf9*-baculovirus expression system vector pAcSG2 (BD Biosciences, San Jose, CA). The light chains (LC), Cdc4 and Rlc1, were cloned into pAcUW51 (BD Biosciences), a dual-promoter vector that drives expression of both LCs. DNA encoding the Myo2 heavy chain was followed by a biotin tag enabling the specific attachment to Neutravidin-labeled microspheres, and a FLAG tag to facilitate purification by affinity chromatography. The biotin tag is an 88-amino acid sequence segment from the *E.coli* biotin carboxyl carrier protein, which is biotinylated at a single Lys residue when expressed in Sf9 cells[54, 55]. Sf9 cells were co-infected with recombinant baculovirus coding for the HC and LC constructs and grown in suspension and harvested at 72 h. The cells were resuspended in ice-cold Buffer F and lysed by sonication and pelleted after addition of 2 mM MgATP. The resulting supernatant was incubated with anti-FLAG resin (Sigma) for 1 h and washed with lysis buffer. The sample was eluted off the column with 100 μg ml$^{-1}$ FLAG peptide (Sigma) in lysis buffer. Protein-rich fractions were pooled and concentrated by Amicon-Ultra filtration (EMD Millipore, Billerica, MA), followed by dialysis against Buffer G.

Catalytically inactive Myo2 (referred to as NEM-Myo2) was prepared by treating 3 μM Myo2 with 1 mM N-Ethylmaleimide (Sigma) in Buffer H for 5 h on ice. The reaction was diluted 10-fold in Buffer I before dialysis against 500 ml Buffer I containing 50% glycerol and storage at −20 °C. Inactivation was tested in a conventional filament-gliding assay, where NEM-Myo2 was non-specifically adhered to the surface of a clean plain coverslip after which phalloidin-stabilized green (ex:488) F-actin (15% Alexa488-actin) in motility buffer (Buffer I containing 2 mM ATP, 15 mM glucose, 20 μg ml$^{-1}$ catalase, and 100 μg ml$^{-1}$ glucose-oxidase) was introduced to the flow chamber and filament-gliding efficiency was assessed by TIRF microscopy.

Amino- and carboxy-terminal SNAP-tagged fission yeast formin Cdc12 (FH1FH2) and mammalian formin mDia2(FH1FH2-C) containing a carboxy-terminal His(6x)-tag to facilitate purification were expressed from an overnight culture of *E.coli* (BL21-Codon Plus (DE3)-RP strain) cells induced with 500 μM IPTG at 16 °C and purified using Talon® metal affinity resin (Clontech, Mountain View, CA)[13]. Chimeric Cdc12(FH1)-mDia2(FH2-C) and mDia2(FH1)-Cdc12 (FH2) formin constructs were cloned by fusing the SNAP-FH1 portion of wild-type formin SNAP-Cdc12(FH1FH2)-6xHis (Met1-Lys287) and SNAP-mDia2(FH1FH2-C)-6xHis (Met1-Phe286) to the respective FH2-6xHis portion of the other formin. Chimeric formin constructs were expressed and purified, as described above.

SNAP-tagged formin constructs were biotinylated and fluorescently labeled overnight at 4 °C using equimolar concentrations of SNAP-Surface™ Biotin and SNAP-Surface™ 549 according to the manufacturer's instructions (New England Biolabs, Ipswich, MA). Excess dye was removed by dialyzing the labeled protein in Buffer E for 6 h at 4 °C[51].

Fission yeast profilin was overexpressed in *E.coli* and purified by poly-L-proline affinity chromatography[56, 57].

**TIRF microscopy**. Ultraclean microscope cover glasses (24 × 40 mm, Fisher Scientific, Waltham, MA) were prepared, coated with mPEG-silane (5,000 MW Laysan Bio Inc., Arab, AL), and flow chambers (20 × 4 mm) were built using double-stick tape[51].

Biomimetic nodes containing myosin Myo2 or formin were prepared by coating non-fluorescent Neutravidin-labeled FluoSpheres (diameter 1 μm, Life Technologies) with biotinylated red (SNAP549) formin constructs or Biotin-Myo2. In brief, 10 μl microspheres (∼ 1.8×10$^8$ particles) was washed with ddH$_2$O and subsequently incubated with 5 μM formin or 3 μg ml$^{-1}$ Myo2 in 30 μl Buffer D for 60 min at 4 °C. Microspheres were then washed three times with Buffer D (plus 1% BSA), recovered in 30 μl Buffer D (plus 0.1% BSA) and stored on ice. Similar to what has been proposed for in vivo node attachment[4, 12], formins and myosins were attached to their amino- and carboxy-terminal ends, respectively.

Immediately before each experiment, myosin- and formin-coated microsphere stocks were diluted 2- to 5-fold into Buffer D (containing 0.1% BSA) yielding appropriate microsphere densities. Microspheres were incubated for 3 min before the flow chamber was rinsed with Buffer D (plus 1% BSA) and incubated for an additional 2 min. Immediately before the actin polymerization mix was applied, the flow chamber was rinsed with a 1:1 dilution of Buffer A. At last, 1.5 μM MgATP G-actin (10–15% Oregon green- or Alexa488-actin) was mixed with 3 μM profilin in Buffer A and immediately transferred to the flow chamber.

In cases where individual red SNAP-formin dimers were immobilized on the glass surface, ultraclean PEG-silane coated glass coverslips were passivated with streptavidin (0.5 mg ml$^{-1}$ in water), incubated with biotinylated SNAP-formin construct, and blocked with Buffer E (containing 0.5% BSA). Myosin-coated microspheres were included following the procedure described above.

TIRF microscopy images of Oregon green- or Alexa488-labeled actin (ex:488 nm), and SNAP-549(red) formin (ex:561 nm) were collected at 5 s intervals with an iXon plus X-4818 EMCDD camera (Andor Technology) using an Olympus IX-50 microscope equipped with a plan-apochromatic through-the-objective TIRFM illumination lens (100 ×, N.A. 1.45).

**Quantitative immunoblotting of Myo2 on microspheres.** The number of Myo2 motor heads bound to the microspheres was analyzed by quantitative immuno-blotting with an antibody against the amino-terminal FLAG-tag epitope (DYKDDDDK) of the Myo2 head domain. Myo2-coated biomimetic nodes were prepared as described above, where for this purpose microspheres were coated with 30 μl containing initial concentrations of either 3 μg ml$^{-1}$ (corresponds to the condition used in all experiments), 1.5, 0.75 or 0.375 μg ml$^{-1}$ myosin. Microsphere density was monitored over the course of the preparation procedure yielding 1.35×10$^8$ particles per 30 μl sample. Samples were separated by SDS–PAGE (7.5% bis-acrylamide) and transferred onto a PVDF membrane (Immobilon-FL, 0.45 μm, Millipore) using a semi-dry transfer apparatus. Biomimetic node-bound myosin was detected by immunoblotting with 1:500 diluted mouse anti-FLAG M2 primary antibody (monoclonal, Sigma, #F3165) and 1:5,000 diluted IRDye 680RD-conjugated goat anti-mouse IgG (H+L, polyclonal, Licor Biotechnologies, Lincoln, NE, #P/N 925-68070) secondary antibody. Immunoblotted samples of-interest were imaged at 700 nm wavelength and quantified by densitometry using a Myo2-standard of known concentrations (indicated in Supplementary Fig. 2b) of the 170 kDa full-length product. The total number of myosin heads per sample was determined and divided by the number of microsphere particles (9×10$^7$ particles) to calculate the motor head density. For Myo2-coated beads that were used in in this study, there are 2,000 myosin heads per bead yielding a density of 637 myosin heads per μm$^2$.

**S. pombe strains.** The fission yeast strains used in this study are listed in Supplementary Table 3. N-terminal *cdc12* (residues 1–881) was amplified by PCR (iProof, Bio-Rad Laboratories, Hercules, CA) from *S. pombe* genomic DNA and cloned into pBluescript II KS(-) (Stratagene) with restriction enzymes XhoI and BamHI. mDia2 FH1FH2 domains (residues 521–1034) were fused to C-terminal *cdc12* (residues 1391–1841) by overlap PCR, and cloned into pBluescript-*cdc12*(N) by homologous recombination using the In-Fusion Advantage PCR Cloning Kit (Clontech) to make *cdc12(N)::mDia2(FH1FH2)::cdc12(C)*. The *cdc12* promoter (1–700 bp upstream of the translation start site) (SacI) and monomeric GFP (NotI to SalI) were amplified and cloned into the *S. pombe* integration vector pJK210[58]. The *cdc12* chimera construct was cloned by In-Fusion (XhoI to NotI) into the pJK210 vector to generate a plasmid containing the *cdc12* promoter and *cdc12(N)::mDia2(FH1FH2)::cdc12(C)::mGFP*. Inserts of the recombinant plasmids were confirmed by sequencing.

The chimera formin construct (*pJK210-P$_{cdc12}$-cdc12(N-term)-mDia2(FH1FH2)-cdc12(C-term)-GFP::ura4+*) was integrated into the *ura4* locus[58]. Endogenous *cdc12* was deleted through *Kan*-cassette gene replacement[59]. Markers for contractile rings, *rlc1-tdTomato-NatMX6*[60], and spindle pole bodies, *sad1-tdTomato-NatMX6*[61], were introduced to the formin chimera strains by mating.

**Fluorescence live cell microscopy and imaging conditions.** Cells were grown in liquid YE5S (yeast extract plus five supplements) media for 20 h at 25 °C and transferred to EMM5S (Edinburgh minimal medium plus five supplements) for 20 h before imaging. Contractile ring assembly was followed in cells spread onto a 25% gelatin EMM5S pad containing 0.1 mM n-propyl gallate[62, 63]. For visualization of cytokinesis nodes, four Z-stacks of 0.5 μm slices were acquired at 100 ms exposure time every 15 s for 20 min. Identical imaging conditions were used for visualization of spindle pole bodies and contractile ring assembly, except when Z-stacks were collected every 20 s.

Spinning disk confocal images were collected with an inverted Nikon Eclipse Ti-E microscope equipped with a CFI Plan Apo 1.2-numerical aperture (NA)/60× water-immersion objective (Nikon, Tokyo, Japan) and a TI-ND6-PFS Perfect Focus unit using 488 nm (*Rlc1-3GFP*) and 561 nm (*Sad1-tdTomato*) illumination from 50 mW solid-state sapphire lasers (Coherent, Santa Clara, CA). Images were collected on an Andor iXon 897 EMCCD camera (Andor, South Windsor, CT).

**BODIPY-phallicidin cell staining and imaging.** Fission yeast cells were stained with BODIPY-phallicidin[64]. In brief, BODIPY-phallicidin powder (Thermo Fisher Scientific, Waltham, MA) was resuspended in methanol to 0.2 units per μl and then aliquoted and lyophilized in a centrifugal evaporator for storage at −20 °C. Cells grown in YE5S were fixed in 16% formaldehyde for 5 min and washed with PEM buffer (0.1 M NaPIPES pH 6.8, 1 mM EGTA, 1 mM MgCl$_2$) three times, both at room temperature, then permeabilized in PEM buffer with 1% triton X-100 for 1 min. After the cells were spun at 7,000 RPM for 30 s, the supernatant was removed and cells were washed in PEM buffer three times. Cells were then resuspended in 10 μl PEM buffer. BODIPY-phallicidin powder was resuspended in PEM buffer to 1 unit per μl and then 1 μl of resuspended phallicidin was added to 10 μl cells and incubated for 30 min in the dark at room temperature. After incubations, cells were washed once with 1 ml PEM buffer, spun at 7,000 r.p.m. for 30 s, and the supernatant was removed. Cells were imaged on glass slides using a Zeiss Axiovert 200 M fitted with a 100×, 1.4 NA objective and Yokogawa CSU-10 spinning disk unit (McBain, Simi Valley, CA) equipped with a Cascade 512B EM-CCD camera (Photometrics, Tuscon, AZ) and a 50 mW 473-nm DPS laser.

**Comparative FH1 domain sequence analysis.** The sequences of formin homology 1 (FH1) domains from *S.pombe* Cdc12 (accession number: CAA92232.1) and *M. musculus* mDia2 (accession number: Q9Z207.1) were aligned using MegAlign software (version 14.1.0.118) by DNASTAR, Inc. (Madison, WI). We defined a profilin-binding poly-L-proline track (PBT) as more than three consecutive proline residues[65, 66]. PBT with > 12 consecutive prolines were counted as multiple tracks. Sequence similarity was determined in ClustalW mode, using the blosum62 amino acid table.

**Data quantification and statistical analysis.** All statistical analysis was performed with GraphPad Prism (version 6.0d, GraphPad Software, Inc., La Jolla, CA). Formin-mediated F-actin elongation was quantified by measuring the lengths over time of all filaments that underwent Search-Capture-Pull events. Filament lengths were measured every frame (frame interval 5 s) using ImageJ64 (NIH, Bethesda, MD) for up to 150 frames before and after Capture-Pull, and traces were recorded as regions of interest (ROIs). Plots of length vs. time for each individual filament gave the average elongation rate (subunits s$^{-1}$) before, during, and after Capture-Pull. Using ImageJ64 along with the recorded ROIs, kymographs of representative Search-Capture-Pull events were generated that show the filament length (*y*-axis) over time (*x*-axis) with the barbed ends aligned at the bottom. An event was counted as Capture-Pull when the Myo2 bead of-interest underwent binding to a formin-elongating actin filament for longer than three consecutive frames and at the same time showed an obvious displacement towards the formin-bound barbed end of the filament during that encounter. Potential events for which the elongation rate after Capture-Pull (i.e. dissociation of the Myo2 bead) did not resume to the approximate baseline pre-capture elongation rate were excluded from the data set.

For comparison of average formin-mediated F-actin elongation rates, the rates from during and after Capture-Pull were normalized to the rate before Capture-Pull and summarized in Box-Whisker plots (whiskers mark inter-quartile range). Statistically significant differences between the three states were calculated by using an ordinary one-way ANOVA along with the Tukey's multiple comparisons test. Significant differences when $p \geq 0.05$. n.s. indicates non-significant differences. A minimum of 7 independent but in most cases $\geq 10$ independent experiments yielding an obvious Search-Capture-Pull event were used for the analysis (see also Supplementary Table 1). The minimum sample size requirement is based on experience and the fact that the observed differences between different conditions were statistically highly significant ($p \ll 0.05$, Figs. 1e, 2d, 4c, 7b) and never at the borderline value of $p = 0.05$. Importantly, one Search-Capture-Pull event can and often times does include more than one Capture-Pull event.

To estimate the drag forces that are at play during myosin-mediated pulling on formin-bound actin filaments in our in vitro setup, we have computed the mean-square-displacement (MSD) value of the myosin-coated bead prior Capture-Pull in $n = 76$ different experiments. The MSD values were determined with $MSD(t) = \langle (x(t)-x(0))^2 \rangle$, with $x$ being the position of the microsphere at time $t$ by tracking the center of each microsphere frame-by-frame (frame interval 5 s) for at least 20 frames. In those cases where the beads exhibit a diffusive-type behavior, we were able to deduce the drag coefficient $\zeta$, viscosity $\eta$ and diffusion coefficient $D$ using the Stokes-Einstein relation. By fitting MSD vs. $t^\alpha$ we determined whether the bead motion was diffusive or sub-diffusive. We found that $n = 18$ were diffusing ($0.8 < \alpha \leq 1.05$) and $n = 58$ were sub-diffusing ($\alpha < 0.8$) (Supplementary Fig. 10a). To extract the diffusion coefficient, we then fit MSD vs. $4Dt$. In the case of diffusive bead motion, assuming a 1-micron diameter, fitting the average MSD vs. $4Dt$ yielded a diffusion coefficient of $D = 3.86 \times 10^5$ nm$^2$ s$^{-1}$, while the median value fitting each experiment separately was $D = 1.30 \times 10^5$ nm$^2$ s$^{-1}$. Given that fitting the average curve is likely to be more accurate than using the MSD for individual trajectories, we chose to estimate the viscosity of the medium coefficient using the average data. The drag coefficient obtained in this manner is $\zeta = k_B T/D = 1.10 \times 10^{-4}$ pN s nm$^{-1}$. The Stokes–Einstein relation gives the viscosity as $\eta = k_B T/(6\pi Dr) = 0.012$ Pa s, where $r = 500$ nm reflects the bead radius. This is in good agreement with the theoretical dynamic viscosity of an aqueous solution containing 0.8% methylcellulose (Supplementary Fig. 5a, b). However, it is

clear in Supplementary Fig. 10a that a large number of beads are either diffusing more slowly or moving less than it would be expected from viscosity $\eta = 0.035$ Pa s. Hence, we posit that those beads would interact with the cover slip and thus produce larger drag forces on the beads. We account for this in our computational model by allowing the beads to experience higher viscosities than the filaments, which are presumably experiencing the average methylcellulose solution viscosity. To get an estimate for what values to use in the modeling, we fit MSD vs. $4Dt$ for all the events to get approximate viscosities that would result in curves close to the sub-diffusive traces in Supplementary Fig. 10a. The median value, which derives from the range of curves above and below the reference lines in Supplementary Fig. 10a, yields a drag value of $\zeta = 2.1 \times 10^{-3}$ pN s nm$^{-1}$ and viscosity $\eta = 0.22$ Pa s, which is ~ 20-times higher than the theoretical solution drag and viscosity.

To determine the velocity of biomimetic node coalescence, kymographs of the Search-Capture-Pull events were used. The Neutravidin-labeled microspheres auto-fluoresce, allowing us to trace the translocation of the myosin-coated microspheres over time to calculate their net velocities. As the kymographs were generated by barbed filament end alignment with the formin-coated microsphere fixed, the determined velocities (Supplementary Table 2) actually represent the sum of both the formin- and myosin-microsphere velocity.

Filament tension $T$ was determined by calculating the ratio of contour filament length (from node-to-node or formin-to-myosin bead) and biomimetic node-to-node (or formin-to-myosin bead) distance at the point of capture. Values $T = 1$ and $T > 1$ represent filaments that are either completely pulled tight or contain some slack, allowing us to distinguish filaments under high ($1 \leq T \leq 1.15$) and low tension ($T > 1.15$).

To visualize node distribution in *cdc12* control and *mDia2* mutant *S. pombe* cells during contractile ring assembly, 3-dimensional surface plots were generated using open source software Gnuplot (http://gnuplot.info). Pixel intensities were averaged using the individual intensity values for each of the eight surrounding pixels, the average background intensity values were subtracted. The arbitrary units were scaled so that the peak intensity for the respective cell was 1.0, making control *cdc12* and mutant *mDia2* 3D projections comparable. Representative cells expressing the ring marker Rlc1-3xGFP were selected and 3D surface intensity plots were generated from still images taken 8 min before a mature ring had assembled. Cell dimensions were plotted along the x- and y-axis in increments of 2 μm, while relative fluorescence intensity (in a.u.) was plotted along the z-axis.

Contractile rings in fission yeast cells were visualized upon BODIPY-phallicidin staining. An ROI encompassing the contractile ring was created in cells in which the contractile ring had formed but not yet begun constriction. Mean fluorescence of the BODIPY-phallicidin in each ring was measured using ImageJ64. In line with previous studies that performed these types of analysis[64], a minimum of 20 randomly picked cells were analyzed and unless specified otherwise the two groups were compared using the Student's *t*-test (unpaired, one-sided).

**Full details of mathematical modeling**. *Determining Myo2 head number engaged with captured F-actin*: To compute how many myosin heads are able to bind to an actin filament, we compute the fraction of the surface area within a certain distance of the actin filament. For this calculation, based on the fact that the entire surface of the microspheres is coated with Neutravidin, the ~ 2,000 myosin heads were treated as uniformly distributed across the surface area of the myosin bead (radius $R_{bead} = 500$ nm) in the in vitro experiments and 20 myosin heads[12] are spread evenly over half the surface of a spherical node (radius $R_{node} = 25$ nm) in vivo (Supplementary Fig. 2d). Therefore, the myosin head density in vitro is 637 per μm$^2$ in vitro and ~ 5,000 per μm$^2$ in vivo.

To account for some flexibility of an individual myosin, we make the approximation that the ability to bind at a distance is Gaussian distributed around a mean length of $l_{myo} = 100$ nm[67] with a s.d. of 5%, that is $\sigma_{myo} = 0.05\, l_{myo}$:

$$G(x) = e^{-\frac{(x - l_{myo})^2}{2\sigma_{myo}^2}}$$

Due to the (assumed) spherical symmetry of the nodes, we assume without loss of generality that in cartesian coordinates the node is aligned with $x = 0$, and separated by a distance $d_{actin}$ in the z-direction and that the myosin binds to the nearest position along the filament (at the same y position as the myosin is bound to the sphere). In spherical coordinates, this assumption can be represented as:

$$D(d_{actin}, R, \theta, \phi) = \sqrt{\delta x^2 + \delta y^2 + \delta z^2}$$
$$= \sqrt{(R\sin(\theta)\cos(\phi))^2 + 0^2 + (R + d_{actin} - R\cos(\theta))^2}$$

Given these assumptions, we integrate over the surface in spherical coordinates "counting" the number of myosins within range to bind the actin filament as a function of the distance of the actin filament from the bead/node.

$$N_{bead}(d_{actin}) = \frac{N_{myo} \int_0^\pi \int_0^{2\pi} G(D(d_{actin}, R_{bead}, \theta, \phi)) \sin(\theta)\,d\theta\,d\phi}{4\pi}$$

The results of the calculation for these parameters are found in Supplementary Fig. 2d showing that for the number of myosin heads on the bead calculated from Supplementary Fig. 2b (2,000) and the number on a node (20) reported in ref. [12], the number of myosin heads that can engage with an actin filament when the filament is separated from the bead by the size of the myosin is similar. This seems reasonable, given our observations that the myosin beads behave processively and

show relatively long residence times (tens of seconds, e.g., Supplementary Movie 2) on actin filaments once they are engaged.

*Modified search-capture-pull model*: To study the forces and dynamics of bead-bead coalescence based on the previous works of Vavylonis and co-workers[3, 27], we designed a simplified Search-Capture-Pull model (all parameters and variables are listed in Supplementary Table 3). The primary difference from the model of Laporte et al.[27] is that we use a representation where the average behavior of motors on the myosin bead is represented in such a way that the motor speed was set rather than the force, prescribing a force-velocity relationship to the myosin behavior. The details of the model are presented below.

Our system has three components: (1) "nodes" of diameter $d$ representing the formin and myosin-coated beads in experiments; (2) actin filaments, represented as polymers that are attached to a formin node and extend from the barbed end; and (3) motor attachments, represented as springs bound at one end to a formin node and at the other end to an actin filament, while processing towards the barbed end of the actin filament.

Simulations were performed in two dimensions for simplicity and to mimic experimental and in vivo conditions, where nodes are constrained to a quasi-2D region. Simulations were started with an initial distance of 5 μm for every data point shown, 48 simulations were run and analysed if a capture event took place, as happened in >50% of cases.

Our system was simulated using overdamped Langevin dynamics as in ref. [27], using the equations of ref. [68] to perform the integration. For each time-step of length $dt$ the positions of every particle $R_i(t)$ was updated using the following rule (with the exception of the motor attachment, see below):

$$R_i(t + dt) = R_i(t) + dt\frac{F_i}{\zeta_i} + \sqrt{\frac{k_B T\, dt}{2\zeta_i}}(W(t + dt) + W(t))$$

where $F_i$ is the force on particle $I$, $\zeta_i$ is the drag felt by particle $I$, and $W(t)$ is a random number chosen at time $t$ from a Gaussian distribution of mean = 0 and unit standard deviation.

The forces in the above equation come from the derivative of the energy function of the system. There are three types of interactions in the system. An actin filament is composed of a series of particles connected by springs, and a motor attachment is also a spring, each having a rest length $l_0$ and spring constant $k$, such that

$$U(l) = \frac{1}{2}k(l - l_0)^2$$

For actin filaments, we chose a value of $k$ smaller than the actual measured value as in ref. [27] for efficient simulations, which does not seem to affect any important properties of filaments or even cross-linked actin networks. Actin filaments have an angular potential of the form

$$U(\theta) = \frac{1}{2}\kappa(\theta - \theta_0)^2$$

where $\kappa = k_B T L_p/l_0$ was set to achieve an actin persistence length value $L_p$, with $l_0$ the length of an actin segment, and $\theta_0$ setting the rest angle to straight (180°). Finally, as in ref. [27] a restoring torque was applied using the first two particles of the actin filament to keep the filament at its initial angle from the formin bead, and in contrast to ref. [27], we do not allow this angle to change since it is also not observed within the resolution of our experimental setup. This force is applied perpendicular to the angle between the filament and the formin bead, and is of the form

$$F^{rot}(\theta) = \frac{k_{rot}}{l_0}(\theta - \theta_0)$$

where $l_0$ is the length of the first actin segment. A force of this magnitude is applied to the first particle of the actin filament and the negative of that force to the second particle of the filament (with overall sign depending on the definition of vector perpendicular to the filament).

Drags per particle $I$, $\zeta_i$ are set as in ref. [27] and are proportional to the viscosity, which can be tuned depending on the experimental conditions. In the Langevin dynamics described above, nodes by default feel a drag viscosity $\zeta_i = 6\pi\eta(D/2)$ and actin particles feel a drag for a rod length $l_0$:

$$\zeta_i = \frac{4\pi\eta l_0}{\ln\left(\frac{l_0}{2a}\right) + 0.84}$$

where $a$ is the radius of an actin filament. We also allowed the drag on the nodes to be set independently of the drag on the actin filaments, to allow us to test the observed effect of variable experimental conditions resulting in different drag forces on biomimetic nodes in vitro, and the effect of being membrane-bound in vivo.

*Actin filaments*: For the data analysed in this work, each formin node has two actin filaments attached and polymerizing, which balances the polymerization force before each capture. In each case, a random angle to the horizontal was chosen between −60° and 60°. The first actin segment connects the center of the formin bead to the first actin particle, which is placed on the surface of the formin node to make this initial angle. A second actin particle is added $l_0$ away from the first actin

particle and the formin node center. The spring connecting the formin node and first actin particle has the same stiffness as the actin segments, scaled linearly to account for its length of $D/2$. A second filament is initialized diametrically opposite to this one.

Polymerization is accomplished by extending the segment between actin particle 1 and 2 from length $l_0$ to $2l_0$ linearly. When this segment reaches twice its rest length, a new particle is inserted at the average position of these two particles. The spring constant on this first segment was scaled linearly to account for the longer length. In order to study the effect of formin arrest on node behavior, we set out to determine the force on the formin, and we chose to define this as the extensional force on this first actin segment.

$$F_{\text{formin}} = k(t)(l(t) - l_0(t))$$

where the spring constant, length and rest length all change with time due to the extension of this segment. When the formin force exceeded a threshold $F_{\text{cut}}$ in any given time step, the extension of the first segment was stopped.

*Myosin dynamics*: Myosin behavior is all performed at a longer time scale ($t_{\text{event}}$) than the molecular dynamics event time, both for efficiency and to allow the system to respond to discrete myosin events, as would occur under experimental in vivo conditions. The distance of the closest particle on each actin filament to the myosin node was computed every $t_{\text{event}}$, and when the closest particle on the actin filament fell within the capture radius (chosen as 110% of the node radius) a myosin link was added between the myosin bead and the closest particle on the actin filament. For simplicity in interpreting our results, we chose not to use a finite attachment rate, and neither did allow for detachment of the myosin node once engaged with the actin filament.

The myosin attachment is represented as a spring, and the force due to stretching was used with a linear force–velocity relationship to determine the myosin 'walking' speed between 0 and $2\,v_{\text{myo}}$.

$$v_{\text{myo}} = v_{\text{myo}}^0 \left(1 - \frac{F}{F_{\text{cut}}}\right), \; |F| < F_{\text{cut}}$$

The myosin attachment on the actin filament has a fixed relative position between the two actin particles of the segment to which it is attached, such that normal dynamics of the system stretches or compresses the motor as a regular spring. Every $t_{\text{event}}$, the position of the motor head is moved towards the barbed end by a distance $d = v_{\text{myo}} t_{\text{event}}$. The forces on the myosin shown in Supplementary Figs. 5 and 8 are computed directly from the extension of myosin's internal spring.

**Code availability**. The simulation code used in this study is freely available for use and modification upon request to the authors.

**Data availability**. All relevant data are available from the authors upon reasonable request.

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

## Acknowledgements

We thank Matthew Lord for helping to conceptualize this project. We thank Jané Kondev (Brandeis University) for extremely helpful discussions concerning force propagation mechanisms through entropic springs. We also thank members of the Kovar and Trybus labs for reagents and helpful discussions. This research was supported by DOD/ARO through a MURI grant, number W911NF1410403, on which G.A.V., E.M.D.L.C. and D.R.K. are co-investigators. This work was also supported by a DFG (German Research Foundation) Postdoctoral Research Fellowship Zi 1496/2-1 (to D.Z.), NIH Molecular and Cellular Biology Training grant T32 GM007183 (to K.E.H.), NIH Ruth L. Kirschstein NRSA F32 GM113415-01 (to G.M.H.), and R01 GM078097 to (K.M.T.).

## Author contributions

D.Z. and D.R.K. designed experiments and wrote the paper (with input from G.M.H. and K.M.T.). D.Z. and K.E.H. performed the experiments and data analysis. G.M.H. and G.A.V. performed theoretical modeling (with input from E.M.D.L.C.). L.W.P. and K.M.T. expressed and purified Myo2.

## Additional information

**Competing interests:** The authors declare no competing financial interests.

