## [Peer Review File · Nature Communications]

Reviewers' comments:

Reviewer #1 (Remarks to the Author):

Experiments are described in which formin Cdc12 grows actin filaments from a Cdc12-coated bead, and myosin-II (Myo2p) on a second Myo2p-coated bead sometimes binds a filament. The beads then move toward each other with reported speeds 102 ± 17 nm/s. Imaging filaments, the authors report that the actin filament elongation rate is reduced 5-fold when bound by myosin, and conclude that the formin is mechanosensitive, its elongation rate being lowered by the myosin pulling force.

Other experiments include (1) substituting myosin with chemically inactivated NEM-Myo2 that captures but does not pull on actin filaments – no elongation rate reduction is seen; (2) Using mammalian formin mDia2 – no elongation rate reduction; (3) substituting formin Cdc12p with chimera constructs containing swapped FH1 domains of Cdc12 and mDia2 - the construct containing the FH1 domain of Cdc12 had lowered elongation rate when pulled by myosin, but not the other construct. The authors conclude that myosin pulling triggers FH1 domain-dependent formin Cdc12 mechanosensitivity.

A mathematical model is presented that describes the myosin-mediated pulling together of the beads seen experimentally.

An in vivo experiment is described, in which ring assembly is studied in a fission yeast strain with the FH1 and FH2 domains of Cdc12 replaced by those of mDia2. Nodes aggregate into clumps, delaying ring assembly by ~ 7 minutes.

General comments.

These are very interesting experiments and the overall goal, to study the interaction of formin-mediated actin filament growth and myosin-mediated actin filament pulling, is important. Previous in vitro experiments have studied the mechanosensitivity of formin-mediated growth, or the capture and pulling of actin filaments by myosin, but not the two together. This interaction is relevant to assembly of the fission yeast cytokinetic ring, where protein complexes ("nodes") are drawn together in this way.

However the quantitative interpretation of the experiments appears to have major inconsistencies, and the conclusions on formin mechanosensitivity appear incorrect.

Specific points.

1. From the reported microsphere relative speed of ~ 100 nm/s (p.3), medium viscosity $\eta = 0.026$ Pa s (p.13), and bead diameter 1 micron, the pulling force on each bead (assumed identical) is ~ 0.013 pN using Stokes' formula. Immediately from Newton's 3rd law, this is the tension in the actin filament, and this is the force pulling the formin away from the barbed end of the actin filament.

Accepting the reported ~ 5 -fold reduction in elongation rate, this implies a mechanosensitivity of ~ 350 per pN, about 700 times that of mDia1 reported by Jegou et al, 2013 (and of opposite sign). This vast value seems implausible to this reviewer for many reasons, and this value is very different from the value claimed by the authors in the manuscript.

2. Somehow, higher pulling forces of 0.5 pN are claimed on the basis of a model (p. 5). I have no idea how this is arrived at. The model incorporates viscous microsphere drags (the value of viscosity used is actually not stated, as far as I could see, but presumably $\eta = 0.026$ Pa s was used), and the spheres move with about the right speed (red lines, extended data Fig. 5), and yet the forces arrived at are ~ 40 -fold too big.

(Elsewhere, other values are quoted for the drag forces: 0.2 pN (Extended Data Fig 6a), and 0.001-0.5 pN (p.6). Neither are these values consistent with Stokes' law, nor with one another).

3. The filament tension of $\sim .03$ pN means this is the force the myosin exerts. This is way below any stall force. Thus the myosin, apparently, operates under essentially load free conditions and the beads are pulled together at the load free myosin velocity.

This is consistent with the load free velocity data reported by Stark et al., Mol. Biol. Cell, 2010. They find this velocity depends on the number of Myo2 heads pulling each filament. Using the number of engaged heads estimated by the present authors (~ 10 , p.3), the fitting curve of Fig. 8D of that ref gives a load-free velocity ~ 100 nm/s. Thus I suppose one expects the beads to move together at ~ 100 nm/s less the force-free formin-mediated polymerization rate, giving a speed not far off the observed speed (the agreement will be closer if the assumed number of engaged myosin heads is higher).

The authors assert a Myo2 load free velocity of ~ 500 nm/s for this situation (p. 3). If one accepts their 10 heads estimate, and the data of Stark et al, this is incorrect. Presumably the 500 nm/s measured in the gliding assay in the present study was under saturating conditions, an interpretation consistent with Stark et al's reported saturating speed of 535 nm/s.

4. If this interpretation is correct (formins experience tiny forces) why is the elongation rate substantially lowered following myosin engagement? The evidence for the lowered elongation rate is Fig. 1d (filament length vs time). Now if one interpolates linearly between the blue line ("search") and the green line ("dissociation") the mean elongation rate during "capture and pull" would then be almost the same as before and after this middle episode. This originates in the sudden discontinuous decrease (increase) in apparent filament length immediately after capture (release). This raises the possibility that the visible filament length does not faithfully report the actin polymerization rate which is in reality unaltered, possibly due to bending or other effects. This is total speculation. In any case, these jumps should be discussed.

5. The authors found that a microsphere covered with NEM-Myo2 (which binds to but does not pull actin) translocates away from a Cdc12-covered microsphere at a rate equal to the filament elongation rate prior to and after capture (~ 45 nm/s). Now the formins are subject to a compressive force. But the magnitude is similarly tiny to the previous tensile force, so that ~ 45 nm/s appears to be the force-free formin growth velocity.

In summary, based on what is described in the manuscript it seems the forces involved are very small. This is not resolved, presumably, by the actin-bearing bead having a much higher drag than expected due to the attached actin (the mobility of the other bead will then dominate). Possibly the beads are interacting with the cover slip and experience much higher drag forces as a result?

As far as this reviewer can see, for this method to access higher forces needed to probe formin mechanosensitivity, bigger bead drags are needed. Thus, more methyl cellulose, or attach the beads. (The latter measure would of course detract from the method's relevance to cytokinetic node condensation.)

Ben O'Shaughnessy
Columbia University

Reviewer #2 (Remarks to the Author):

This paper attempts to reconstitute aspects of fission yeast cytokinesis in a cell-free system,

including several major components of the process: the formin Cdc12, myosin II, actin and profilin. The major result is a biophysical one: that tension applied to an actin filament by myosin causes a significant decrease in Cdc12-mediated actin filament elongation. They show that this effect is dependent on myosin motor activity and on the Cdc12 formin (the effect does not occur with another formin, mammalian mDia2). They find that the FH1 domain of Cdc12 is the relevant region of the protein affected by tension (and that the FH1 domain of mDia2 is not affected by tension). They conduct some modeling to test the conclusions. They also do a clever experiment in fission yeast cells, where they swap the FH1 and FH2 domains of Cdc12 for those of mDia2, and find that cytokinesis is compromised.

The work makes several very intriguing findings, that inform both actin biochemistry in general and fission yeast cytokinesis specifically. However, as a whole the paper could use some more care, with several experimental issues that could have been better controlled or documented. In general, a number of details are not discussed or discussed too briefly in the text, especially in the modeling and discussion of the differences between Cdc12 and mDia2 FH1 domain differences. The main issues (elaborated in more detail below) are that:

- 1) The two-bead system provides a somewhat messy demonstration of the tension phenomenon, which could be better displayed by Cdc12 attached to the coverslip.
- 2) One figure testing the modeling (2h) appears to be incomplete.
- 3) A vital test of their cellular data is not present – is there more actin at the cytokinetic ring in their mDia2 chimeric cells?

1) Figure 1B first movie – not very clean at all, and is difficult to see the phenomenon well. It would be better to show the Cdc12 adhered to the coverslip (as they use for the NEM=myosin control experiments in Figure 2). This seems to be a much cleaner system. At the very least, a better example is needed.

2) Some comparative sequence analysis of Cdc12 versus mDia2 FH1 domains would be useful here as a supplementary figure, in addition to more discussion. How many potential profilin-binding modules are present on each? Are there any other suggestive features of interest in these sequences?

3) For the tension experiments, it would be very useful to use one of the existing systems (refs 15 or 16) to calculate forces more directly. Or, they could use a cleaner system such as Cdc12 on the coverslip (as they do a little in Figure 2).

4) Issues with Figure 2h:

a. why are there no “low tension” Cdc12 measurements? These are crucial to determine whether conversion from fast to slow elongation is a step function or a gradual change over a range of increasing tension.

b. mDia2’s response to high tension appears very noisy (individual points spreading from 0.5 to >2 elongation rate units). There is some explanation supplied, but it is not all that convincing.

5) Cell experiments:

a. The hybrid to test would be much more relevant if only the FH1 were swapped between Cdc12 and mDia2.

b. The authors hypothesize that the hybrid protein causes more actin to build up at the nascent cytokinetic ring. Can they quantify actin at the ring to test the hypothesis, perhaps by rhodamine-phalloidin staining of fixed cells? Without this result, it is difficult to distinguish between their hypothesis (that mDia2 causes too much elongation) and simply having a somewhat poorly folded chimeric protein.

6) Lines 173-175, discussing G-actin concentrations, come out of nowhere, and it is unclear what exactly their relevance is as written.

7) Figure 2B – can they have another example where there is a longer search phase?

8) page 3. The elongation rates given in the text (10.6 and 11.4) are different than in Figure 1d and 2 b. Some clarification needed. Also, it would be best to give the mean elongation rates for at least the search filaments (and preferably all of the conditions) in the legends for Figure 1c, 2e and 2f (and the relevant graphs in figure 3). In fact, why are the search filament points normalized individually for these plots? This makes it appear that there is no error at all in these readings. It

would be more useful to have the spread of the elongation rates depicted for each of these points.

Response to referees' comments

We would like to thank Dr. O'Shaughnessy (Rev.#1) for his careful and thoughtful reading of the manuscript, and in particular his ensuring that our modeling and simulations are self-consistent and agree with other known results. We were pleased to read that Rev.#1 finds the presented experiments "very interesting" and the overall goal "very important". We agree with Rev.#1's concerns about the presentation of aspects of our modeling data, which were presented in a manner that was confusing and consequently raised questions about certain assumptions. Therefore, in the revised manuscript we have included additional details and explanations of modeling assumptions, which has improved the clarity. Importantly, we are convinced that our data is consistent with formin Cdc12's inhibition via its FH1 domain by forces that are on the order of 0.1-1 pN. However, we emphasize that the demonstrated inhibition of formin Cdc12 due to applied myosin-generated forces is a hypothesis that, while consistent with our data and modeling, remains to be further tested in future studies using assays where it is possible to do have more direct and quantitative force analyses. In this work, we have prioritized the use of the physiological force generator, myosin Myo2.

Authors' reply to specific concerns raised by Reviewer #1:

1) and 2)

From the reported relative microsphere speed of ~100 nm/s (p.3), medium viscosity $\eta = 0.026$ Pa s (p.13), and bead diameter 1 micron, the pulling force on each bead (assumed identical) is ~ 0.013 pN using Stokes' formula. Immediately from Newton's 3rd law, this is the tension in the actin filament, and this is the force pulling the formin away from the barbed end of the actin filament. Accepting the reported ~ 5-fold reduction in elongation rate, this implies a mechanosensitivity of ~ 350 per pN, about 700 times that of mDia1 reported by Jegou et al, 2013 (and of opposite sign). This vast value seems implausible to this reviewer for many reasons, and this value is very different from the value claimed by the authors in the manuscript.

Somehow, higher pulling forces of 0.5 pN are claimed on the basis of a model (p. 5). I have no idea how this is arrived at. The model incorporates viscous microsphere drags (the value of viscosity used is actually not stated, as far as I could see, but presumably $\eta = 0.026$ Pa s was used), and the spheres move with about the right speed (red lines, extended data Fig. 5), and yet the forces arrived at are ~ 40-fold too big. (Elsewhere, other values are quoted for the drag forces: 0.2 pN (Extended Data Fig 6a), and 0.001-0.5 pN (p.7).

Response: Rev.#1's calculations are based on the viscosity of 0.026 Pa s (reported on p. 13 of the original manuscript). As stated on p. 19 of the revised manuscript, this value (revised as 0.012 Pa s) is based on tracking myosin beads that are clearly diffusing prior to capture (see new Supplementary Fig. 10a and new Supplementary Table 5). We were pleased to see that this viscosity value, which was calculated using the Stokes-Einstein relation and was referred to in reviewer #1's comment, is consistent with the viscosity value reported for a solution containing 0.8% methylcellulose (400 cP). To make this important point clear to the reader, we included the following paragraph into the revised manuscript text (p.9, paragraph 3): "*By analysing the mean square displacements of myosin-coated beads prior to capture, we were able to estimate the drag (i.e. friction) experienced by the beads in our in vitro reconstitution setup. Our analysis revealed that based on the diffusive bead behaviour prior to capture (see Methods for details), events take place at an estimated viscosity of $\eta=0.012$ Pa s, which is in good agreement of the theoretical dynamic viscosity of an aqueous solution containing 0.8% methylcellulose (400 cP) (page 5, SIGMA #M0262 product information).*" Further, on p.19f (paragraph 1) of the Methods we included a more detailed description of how MSDs, diffusion coefficients, drag and viscosity were determined.

However, it is important to mention that a large proportion of the beads were observed to move more slowly than what would be expected using the calculated viscosity of 0.012 Pa s. As predicted in the reviewer's conclusions "Possibly the beads are interacting with the cover slip and experience much higher drag forces as a result?" – this is exactly what we think and we now state this claim in the discussion of the revised manuscript text (p.9, paragraph 3): "*However, in 75% of cases the beads*

Response to referees' comments

showed sub-diffusive behaviour, which we attribute to non-specific interactions of the beads with the coverslip, yielding an estimated effective median viscosity value of $\eta=0.22$ Pa s that is experienced by the beads and is ~ 20 -fold higher than the theoretical viscosity of the medium." Moreover, the MSD data (now provided as new Supplementary Fig. 10a) are based on tracking myosin-coated beads before capture, early in the experiments. We know that in 29 % of cases, either the formin or myosin bead does not move during the "Capture-Pull" phase, further indicating interactions of the beads with the cover slip, perhaps over and above our best 'baseline' estimate.

Although perhaps not clear in our original manuscript, this is also the reason why our simulation model allows us to tune the drag coefficient for the 'nodes' independently of the actin. A conservative estimate of an average bead drag coefficient is 20-times higher than expected from solely the methylcellulose viscosity of 0.012 Pa s (based on new Supplementary Fig. 10a), yielding a corrected median viscosity of ~ 0.2 Pa s ($n=18$, Supplementary Table 5). Therefore, in regard to Rev.#1's remark on Extended Data Fig. 6 (Supplementary Fig. 5 of the revised version), an average force on the formin of ~ 0.1 pN is at play when using this 10-times higher drag value on the beads (Supplementary Table 5, shaded box in Supplementary Fig. 8 indicates deviation). Also, p. 5 of the original manuscript stated a drag force of 0.5 pN, which was a typo, as can be seen by examining extended data figures 6 and 9 (Supplementary Fig. 5 and 8 of the revised version). We have corrected this typo to 0.1 pN, which addresses the reviewer's second point ("Somehow, higher pulling forces of 0.5 pN are claimed on the basis of a model"). To make it crystal-clear to the reader why we use the viscosities that we do, we have added the following new Supplementary Fig. 10a and Supplementary Table 5 as well as the following paragraph as part of the Discussion (p. 10, paragraph 1): "*Using $F=v^*(6\pi r\eta)$, we estimate a force of 0.1 pN ($F=50$ nm s^{-1} * $6*3.14*500$ nm * $0.22*10^{-6}$ pN s nm^{-2}) is required to bring two beads together at an average speed of 100 nm s^{-1} (i.e. with the formin bead moving at 50 nm s^{-1} , Supplementary Table 1), which is in strong agreement with the results obtained from our computational model (Supplementary Fig. 8).*"

Additionally, Rev.#1 commented on the fold-reduction-per-pN of the formin. Our asserted force of ~ 0.1 pN together with the ~ 4 -fold reduction in Cdc12's elongation rate, would result in a 40-fold change per pico-newton or smaller. This is about 13-times greater than what Jegou et al. (2013) reported, but at the same time is much less extreme than the 700-fold rate change suggested by Rev.#1. Moreover, since we did not observe intermediate levels of inhibition for Cdc12, the mechanism of inhibition appears to be sigmoidal. In other words, formin activity is quenched to low levels for (what we believe are) small forces in a faster than linear fashion. This may seem to contrast the findings by Jegou et al. (2013: mDia1) and Courtemanche (2013: Bni1) using hydrodynamic flow, where in the range between 0 and ~ 1 pN mechano-acceleration seems to take place linearly proportional to the applied force (Fig. 3c Jegou et al. Nature Comm. 2013). This could be an interesting indication for a difference between FH2 (mDia1, Bni1) vs. FH1 (Cdc12) domain-dependent formin mechanosensitivity, which suggests future investigations comparing these and other formins in a setup where it is possible to carry out direct and quantitative force analyses, while omitting myosin as the physiological force generator.

3) *The filament tension of $\sim .03$ pN means this is the force the myosin exerts. This is way below any stall force. Thus, the myosin, apparently, operates under essentially load free conditions and the beads are pulled together at the load free myosin velocity. This is consistent with the load free velocity data reported by Stark et al., Mol. Biol. Cell, 2010. They find this velocity depends on the number of Myo2 heads pulling each filament. Using the number of engaged heads estimated by the present authors (~ 10 , p.3), the fitting curve of Fig. 8D of that ref gives a load-free velocity ~ 100 nm/s. Thus, I suppose one expects the beads to move together at ~ 100 nm/s less the force-free formin-mediated polymerization rate, giving a speed not far off the observed speed (the agreement will be closer if the assumed number of engaged myosin heads is higher). The authors assert a Myo2 load free velocity of ~ 500 nm/s for this situation (p. 3). If one accepts their 10 heads estimate, and the data of Stark et al, this is incorrect. Presumably the 500 nm/s measured in the*

Response to referees' comments

gliding assay in the present study was under saturating conditions, an interpretation consistent with Stark et al's reported saturating speed of 535 nm/s.

Response: We thank Rev.#1 for making this point. We fully agree with the reviewer's assertion that in light of and in agreement with Stark et al.'s work (MBoC 2010) the myosin bead in our system operates under essentially load-free conditions and thereby the myosin and formin beads are pulled together at the load-free myosin velocity ($\sim 100 \text{ nm s}^{-1}$, Supplementary Table 2). Therefore, in the revised version of the manuscript we have changed our initial interpretation accordingly (p.10, paragraph 2): *"With relatively small forces ranging from 0.01 to 1 pN (Supplementary Fig.5 and Supplementary Table 5), in most cases the applied force is far from Myo2's reported stall force (2 pN per myosin head) (Vavylonis et al. 2008), arguing that in our in vitro system the myosin beads operate under essentially load-free conditions where the myosin and formin beads are pulled together at the load-free myosin velocity of $\sim 100 \text{ nm s}^{-1}$ (Supplementary Table 2)." In particular, with reference to Stark et al. (MBoC 2010) we have stressed the fact that the observed node pulling speed under load-free conditions strictly depends on the number of engaged myosin heads. Our estimate of ~ 10 myosin heads being engaged at all times during Capture-Pull (Supplementary Fig. 2d) is in strong agreement with the numbers Stark et al. (2010) reported for such myosin speeds (Fig. 8D MBoC 2010). Therefore, in the revised version of the manuscript (p.10, paragraph 2 and caption Supplementary Fig. 2) we now speak of the myosin gliding speed (542 nm s^{-1} , Supplementary Fig. 2a) as "saturated" rather than "unloaded" myosin speed: *"At higher myosin densities, Myo2 velocities can saturate at speeds as high as 542 nm s^{-1} (Supplementary Fig. 2a, compared to 535 nm s^{-1} in Stark et al. 2010)."**

4) *If this interpretation is correct (formins experience tiny forces) why is the elongation rate substantially lowered following myosin engagement? The evidence for the lowered elongation rate is Fig. 1d (filament length vs time). Now if one interpolates linearly between the blue line ("search") and the green line ("dissociation") the mean elongation rate during "capture and pull" would then be almost the same as before and after this middle episode. This originates in the sudden discontinuous decrease (increase) in apparent filament length immediately after capture (release). This raises the possibility that the visible filament length does not faithfully report the actin polymerization rate which is in reality unaltered, possibly due to bending or other effects. This is total speculation. In any case, these jumps should be discussed.*

Response: We think Rev.#1's speculation is unsubstantiated for several reasons: 1) As mentioned previously (p.2, response to point 2), the reported changes in formin elongation rates may take place in a sigmoidal fashion, rather than in a linear fashion. It is therefore critical to carefully trace filament lengths frame by frame so that changes in elongation rate are not missed, which avoids false interpolations (if we understand Rev.#1's implication correctly). Therefore, we feel that interpolating data points from start (Search) to end (Dissociation) of an event does not make sense. 2) Because 61% of all Cdc12 Search-Capture-Pull events reveal a more than 3-fold inhibition in elongation rate, which for 83% of all events lasts longer than 35 seconds. 3) As with any experimental study, there will always be variability in the system (a range of drag forces and inherent variance in formin-mediated elongation). For this exact reason, the claims made in this paper are not based on one specific example but instead are founded on the analysis of numerous Cdc12 Capture-Pull events ($n=33$) from independent sets of experiments ($n=18$) from which all the obtained rates were normalized to the pre-capture (Search) rate of the respective event (see also response to Rev. #2's point 8), which makes comparisons between experiments possible (e.g. Fig. 1e, 2d,g, 3c, 4b,c, 5c etc.). 4) We appreciate the theoretical possibility raised by Rev.#1 that filaments could undergo bending and in such cases may make the faithful reporting of filament lengths difficult. However, in addition to the above mentioned numerous events that were recorded and analyzed for one type of experiment (i.e. pulling on formin that is attached to beads, Fig. 1), we provide new data from two different sets of experiments (i.e. pulling on formin that is fixed to the surface via its N- and C-terminus, new Fig. 2a–d and new Fig.

Response to referees' comments

5a–c) re-establishing our initial claim that pulling on actin-bound formin Cdc12 results in its inhibition via the FH1 domain. We therefore are confident that our measurements faithfully support the claims made in this paper.

5) *The authors found that a microsphere covered with NEM-Myo2 (which binds to but does not pull actin) translocates away from a Cdc12-covered microsphere at a rate equal to the filament elongation rate prior to and after capture (~ 45 nm/s). Now the formins are subject to a compressive force. But the magnitude is similarly tiny to the previous tensile force, so that ~ 45 nm/s appears to be the force-free formin growth velocity.*

Response: The reviewer states that "...the magnitude is similarly tiny to the previous tensile force, so that ~ 45 nm/s appears to be the force-free formin growth velocity," raising the question whether compressive forces on the formin could and perhaps should inhibit formin based on the FH1 mechanism that we are suggesting in this work. However, based on further analysis of experiments using inactive NEM-myosin (Fig. 2e–g), we feel strongly that the forces are not the same in this case. An estimated 45 nm s⁻¹ appears reasonable based on the example NEM-myosin trace in Fig. 2e, f of the manuscript. However, in every NEM-myosin event it appears that the filament-bound NEM-myosin bead exhibits a diffusive-like (non-ballistic) behavior, arguing that an equal and opposite pushing force on the formin is minimal. The contrast between the bead behavior in an active and inactive case can be seen in the attached new Supplementary Fig. 10b vs. 10c (traces of active vs. inactive Myo2 beads during the Capture phase for 3 examples), which clearly demonstrates the contrast between the type of dynamics exhibited by a pulled (active, new Supplementary Fig. 10b) and pushed (inactive, new Supplementary Fig. 10c) myosin bead. While previous work (Kovar et al. 2003 and Kovar et al. 2006) shows that formin polymerization can exert ~1 pN of force, these experiments were done with very short actin filaments, where buckling is costly from a thermodynamics perspective. Importantly, even under those compressive forces, formin Cdc12 does not show signs of inhibition, most likely because the observed mechano-inhibition happens through Cdc12's FH1 rather than FH2 domain. In all of our NEM-myosin events, the filaments of interest were over 10 microns in length, where filaments buckle relatively easily. It therefore seems that the force of polymerization is not sufficient to induce significant (pushing) drift on the myosin bead, and hence we are confident that the forces on the formin are of a much smaller magnitude in this case than in the active myosin case. Further, the discussion of the revised version of the manuscript (p.10, paragraph 4) now contains a paragraph addressing this very important point: *"In addition to the myosin pulling force, compressive forces from the formin 'pushing' on the elongating actin filament could potentially factor into the net force that is applied on the formin and hence may affect its inhibition. Experiments where the diffusive behavior of catalytically inactive NEM-Myo2 beads (incapable of exerting active pulling force) during the capture phase was followed, convincingly show that filament-bound NEM-Myo2 beads exhibit a strictly (sub)diffusive behavior when compared to active NEM-Myo2 bead-mediated Capture-Pull (Supplementary Fig. 10b,c). We are confident that under the conditions tested, the force of formin-mediated actin polymerization is not sufficient to induce significant pushing drift on the myosin bead and concomitantly would be too small to contribute to formin Cdc12's mechano-inhibition under these conditions."*

In summary, based on what is described in the manuscript it seems the forces involved are very small. This is not resolved, presumably, by the actin-bearing bead having a much higher drag than expected due to the attached actin (the mobility of the other bead will then dominate). Possibly the beads are interacting with the cover slip and experience much higher drag forces as a result?

As far as this reviewer can see, for this method to access higher forces needed to probe formin mechanosensitivity, bigger bead drags are needed. Thus, more methyl cellulose, or attach the beads. (The latter measure would of course detract from the method's relevance to cytokinetic node condensation.)

Response to referees' comments

Response: As we mentioned above (response to specific points 1 and 2), we absolutely agree that due to bead-cover slip interactions for a number of events the drag forces at play are much higher than 0.1 pN (in some cases perhaps even exceeding 1 pN, new Supplementary Figure 8, new Supplementary Fig. 10a and new Supplementary Table 5). We have therefore indirectly addressed Rev.#1's point concerning the suggestion to try conditions at higher drag. In theory, it is certainly a fair suggestion to increase the viscosity, however in practice we think it will be a non-trivial experiment because at higher viscosities, among other things, monomeric actin from solution will be depleted by non-specifically adhering to the coverslip, resulting in unwanted elongation rate artifacts.

Instead, we would like to point the reviewer's attention to a more critical new experiment we have included in the revised manuscript, which further substantiates our interpretation that mechano-inhibition maps to the FH1 domain. We have included additional data (new Fig. 5) from a new set of experiments where we fixed formins Cdc12 and mDia2 to the coverslip via their FH2 domains as opposed to their FH1 domains (Figures 1–4). This experiment tests directly our claim that mechano-inhibition of Cdc12 maps to its FH1 domain. As expected, myosin pulling on Cdc12's FH2 domain does not inhibit Cdc12 (and neither mDia2) strongly arguing that only through anchoring formin Cdc12's FH1 domains (as it is thought to occur *in vivo*) to the cell membrane (glass surface *in vitro*) formin Cdc12 can exhibit its mechanosensitivity.

We hope to have adequately addressed Rev.#1's points of concern and would like to once again thank Reviewer #1 for his thoughtful comments, concerns and suggestions through which the quality of the revised manuscript is significantly improved.

Response to referees' comments

We thank Reviewer#2 for the thorough assessment of the manuscript. We were very pleased that Rev.#2 appreciates the significance of our work and found our results intriguing in that they advance the knowledge of actin biochemistry in general and fission yeast cytokinesis specifically. We particularly appreciate the reviewer's suggestions on how to further improve the presentation and discussion of key findings and very much hope to have adequately addressed any of the reviewer's concerns in the revised manuscript. Specifically, we have added additional data sets that include data from the proposed experiment (Specific points 1 and 3) to fix formin Cdc12 directly to the coverslip surface in order to have an alternative and as Rev.#2 put it "cleaner system" than the initially presented attachment of formins to a bead (Fig. 2a–d). Identical to what we found for Cdc12 attached to beads, surface-attached formin Cdc12 undergoes significant inhibition when it is being pulled on by myosin Myo2. This set of new experiments also emphasizes, as initially stated, formin Cdc12 inhibition does not require complete filament tension (Specific Point 4a, new Fig. 6). We further included new in vivo cell data (fluorescent BODIPY-phalloidin images) that further strengthen our hypothesis that *mDia2* fission yeast cells generate excessive ring F-actin, which attributes to node clumping and delay in proper ring assembly (Specific Point 5b, new Fig. 7d,e). Finally, we would like to draw the reviewer's attention to a new figure, which describes new data from experiments where formins Cdc12 and mDia2 were fixed to the coverslip surface via their FH2 domain instead of their FH1 domain (new Fig. 5). In addition to the previously presented in vitro chimera experiments (Figures 1–4), these data provide independent evidence that the FH1 domain of formin Cdc12 comprises the force-sensitive region and plays a key mechanistic role in the mechano-inhibition of formin Cdc12.

Authors' reply to specific concerns raised by Reviewer#2:

1) *Figure 1B first movie – not very clean at all, and is difficult to see the phenomenon well. It would be better to show the Cdc12 adhered to the coverslip (as they use for the NEM=myosin control experiments in Figure 2). This seems to be a much cleaner system. At the very least, a better example is needed.*

Response: We agree with the reviewer in that the movie used in Fig. 1B is perhaps not ideal because there is a significant amount of F-actin generated by picomolar concentrations of formin left over in solution. In the original manuscript, we had included an alternative example (Extended Data Fig. 3) where there is less F-actin in solution. For the revised version, we decided to include a different alternative figure (new Supplementary Fig. 1 and new Supplementary Movie 2), which in our opinion now makes it much easier for the reader to assess the reconstituted Search-Capture-Pull event that is shown. Further, to address the reviewer's suggestion to show an example of surface-adhered Cdc12, we carried out these new experiments and compiled a new figure (Fig. 2a–d) containing an example for in vitro reconstituted Search-Capture-Pull of surface-adhered Cdc12 (Fig. 2b,c). In addition to the new less complicated examples of reconstituted Search-Capture-Pull, we decided to also keep the original example shown in Fig. 1b-d (Supplementary Movie 1) because it is a fantastic example where both the formin- and myosin-beads coalesce toward each other, mimicking in vivo Capture-Pull scenario during ring assembly.

2) *Some comparative sequence analysis of Cdc12 versus mDia2 FH1 domains would be useful here as a supplementary figure, in addition to more discussion. How many potential profilin-binding modules are present on each? Are there any other suggestive features of interest in these sequences?*

Response: We appreciate this reviewer's comment and agree that discussing the potential differences between the FH1 domains of formins Cdc12 and mDia2 as well as pointing out possible reasons that may account for rendering Cdc12 mechano-sensitive while leaving mDia2 mechano-insensitive, will greatly improve the manuscript. We therefore included a new supplementary data figure (Supplementary Fig. 9), showing the suggested comparative sequence analysis of Cdc12 and

Response to referees' comments

mDia2 FH1 domains. Further, we highlighted profilin-binding poly-L-proline tracks (PBT) pointed out by Rev.#2, which makes obvious the differences in the spacing in-between PBTs as well as the distance to the adjacent carboxy-terminal FH2 domain. These are features that we (supported by computational work from Bryant et al. 2017) suggest may play an important role for formin Cdc12's mechanosensitive response and will be the focus of future experimental and computational studies. Further, the discussion of the revised manuscript (p.9, paragraph 1) now contains a detailed paragraph outlining potential FH1 features that could convey Cdc12's mechano-inhibitory behavior. "A comparative sequence analysis of Cdc12 and mDia2 FH1 domains (Supplementary Fig. 9) shows that in addition to a general lack of sequence similarity, Cdc12 FH1 is ~25% shorter and its two profilin-binding proline-rich tracks (PBT1 and PBT2) are considerably farther away from the carboxy-terminal FH2 domain (distance to FH2 domain for PBT1 and PBT2 of Cdc12 vs. mDia2: 60 and 20 vs. 31 and 15 amino-acids). The computational model proposed by Bryant et al. would predict that under force Cdc12 undergoes deceleration and mDia2 acceleration (Fig. 5 Bryant et al. 2017), which is in good agreement with our experimental data (Fig. 1e, 2d and 3c)."

3) For the tension experiments, it would be very useful to use one of the existing systems (refs 15 or 16) to calculate forces more directly. Or, they could use a cleaner system such as Cdc12 on the coverslip (as they do a little in Figure 2).

Response: In light of Rev.#2's concern that force/tension calculations may be less accurate in a two-bead system, we followed the reviewer's suggestion and carried out multiple independent sets of new experiments (n=7 independent sets with 19 individual Capture-Pull events) where formin Cdc12 is now fixed to the coverslip surface instead of a bead. Therefore, in the revised manuscript we have included a new figure, showing a representative example (kymograph and filament elongation over time, new Fig. 2b,c), and Box-Whisker plot of the normalized rate changes before, during and after Capture-Pull (n=19 events, new Fig. 2d). Further, the relationship of filament tension and rate change was included in Fig. 6b,c (previously Fig. 2h). identical to what was initially shown for Cdc12 attached to beads, surface-attached formin Cdc12 undergoes significant inhibition when it is being pulled on by myosin Myo2. Furthermore, as initially stated, formin Cdc12 inhibition does not require complete filament tension as is nicely recapitulated by this new set of experiments where Capture-Pull on Cdc12-bound filaments that are not completely taught still triggers formin inhibition ($T > 1.15$, new Fig. 6).

We refrained from using the previously applied microfluidics approach by Jegou et al. (Nature Comm. 2013) or Courtemanche et al. (PNAS 2013) due to the following two reasons: 1) As was pointed out by Rev.#1, a significant asset of this paper is that the physiological force generator, myosin, is used throughout the paper so that we felt most comfortable to keep the core experimental system (formin vs. myosin) consistent. 2) Based on the computational work (Supplementary Fig. 5 and 8), we estimate sub-piconewton forces (~0.1 pN) to be sufficient to inhibit Cdc12. Such low forces are typically below the reliable resolution limits of microfluidics and therefore may not add additional insight.

4) Issues with Figure 2h:

a. why are there no "low tension" Cdc12 measurements? These are crucial to determine whether conversion from fast to slow elongation is a step function or a gradual change over a range of increasing tension.

Response: We thank the reviewer for this suggestion. We should have discussed better the point that in Fig. 2h of the original manuscript there are no 'low' tension ($T > 1.15$) Cdc12 events. As we stated on p.6 (paragraph 2) of the revised manuscript, we hypothesized "... that inhibition of Cdc12 ensures that filaments experience tension almost instantaneously (new Supplementary Fig. 6), and that relatively low forces (~0.5 pN, now revised to 0.1 pN; new Supplementary Fig. 5a and 8a) are sufficient

Response to referees' comments

to effectively inhibit Cdc12 without dissociating formin from the end.” Specifically, with an average filament contour length of $13.4 \pm 1.7 \mu\text{m}$ (Median \pm S.E.M.) and an average formin-to-myosin (bead) distance of $12.5 \pm 1.4 \mu\text{m}$ (Median \pm S.E.M.), filaments have low slack to begin with. In addition, with an average (pre-capture) filament elongation rate of 9.6 subs s^{-1} ($\approx 35 \text{ nm s}^{-1}$) the average myosin pulling speed (93 nm s^{-1}) is three times higher, thus forcing captured filaments to rapidly straighten out (new Supplementary Fig. 6 and Supplementary Tables 1 and 2). Hence, in order for the captured filament to be not completely straightened out (i.e. ‘low’ filament tension), (a) the rate at which the formin elongates a filament would have to be faster than the myosin-bead pulling speed, and/or (b) the filament length in-between both attachment points (formin and myosin) would have to be significantly larger than the net distance between the two attachment points at capture onset. This explains why for mDia2 (three to four times faster elongation rates than for Cdc12) multiple ‘low’ tension events were recorded (new Fig. 6b,c open grey symbols). The new data set of surface-adhered Cdc12 directly addresses Rev.#2’s concern in that six of the nine Cdc12 inhibition events take place in the ‘low’ filament tension regime (new Fig. 6b,c), directly underscoring our previous statement (p.6, paragraph 2) that “...the forces propagated to filament-bound Cdc12 are largely independent of filament tension (Supplementary Fig. 5)...”. Based on our mathematical model (Fig. 5a and 8a) we think that relatively small forces suffice to inhibit Cdc12, suggesting a step function, rather than a gradual change since this type of mechanoregulation seems not to depend on the degree of filament tension. It should be noted that in vivo the node-to-node distance is very short (600 nm) so that captured formin-bound filaments will always be under relatively high tension upon formin inhibition, ensuring the immediate and proper coalescence of neighboring nodes (Fig. 8).

The discussion of the revised version of the manuscript (p.11, paragraph 2) now contains a paragraph addressing these very important points: “However, even under decreasing filament tension (i.e. increasing filament slack) conditions Cdc12 still undergoes significant inhibition (open red symbols, Fig. 6b,c), demonstrating that Cdc12 mechano-inhibition does not require filament tension. This directly emphasizes the notion that the relatively low forces ($\sim 0.1 \text{ pN}$) propagated to filament-bound Cdc12 are largely independent of filament tension, as is also shown using our mathematical model (Supplementary Fig. 5a and 8a,b). We ascribe this phenomenon to the actin filament behaving as an entropic spring, whereby force from the myosin propagates through the filament via thermal fluctuations of the filament itself and regardless of how extended the filament is” Phillips, Kondev and Theriot Chapter 8 ”

- b.** mDia2’s response to high tension appears very noisy (individual points spreading from 0.5 to >2 elongation rate units). There is some explanation supplied, but it is not all that convincing.

Response: We do agree that the presented data on mDia2’s probable activity enhancement under high tension (previously Fig. 2h, now Fig. 6b,c in the revised manuscript) should be treated only as an initial observation, and therefore should be interpreted appropriately. However, we believe that instead of the mDia2 data being “noisy”, the presented data are indicative of a complex nature that will require an additional study, such as applying defined force ranges by using hydrodynamic flow experiments. Therefore, in the discussion section of the revised version (p.9, paragraph 2) we expanded upon the statement that “...mDia2’s potential mechano-enhancement may also depend on its FH1 domain, however much higher forces and/or filament tension may be required.” In summary, in the revised version we hypothesize that (a) the contribution of mDia2’s FH2 domain, and/or (b) larger forces ($> 1\text{pN}$) could result in mDia2’s ability to respond positively to forces, in which case a different force-sensing mechanism would be at play.

Importantly, we have also added the analysis of new additional experiments where instead of anchoring mDia2 via its FH1 domain (Fig. 3a–c), mDia2 (and Cdc12) were fixed to the coverslip surface via their FH2 domains (new Fig. 5). These new data indicate that the direct pulling of myosin on mDia2’s FH2 domain has no significant impact on mDia2’s activity (new Fig. 5c), suggesting that

Response to referees' comments

either the enhancement of mDia2 activity is dependent upon stretching the FH1 domain, and/or higher forces as well as filament tension are required (new Fig. 6).

Future studies where higher forces can be applied by using hydrodynamic flow will be required to further dissect mDia2's response behavior to forces greater than the physiological myosin-mediated forces investigated in this study.

5) Cell experiments:

- a. *The hybrid to test would be much more relevant if only the FH1 were swapped between Cdc12 and mDia2.*

Response: We had considered the possibility of swapping just the FH1 domains (rather than FH1 and FH2 domains altogether), but ultimately rejected this experiment because we feared that the results will likely be difficult to interpret. As shown in Supplementary Fig. 3 (and Supplementary Table 1), swapping only the FH1 domains significantly alters the actin filament elongation rates, which is likely to cause changes in ring assembly dynamics on its own. Specifically, swapping mDia2's FH1 domain for Cdc12's FH1 domain yields elongation rates that are 5-times faster than those recorded for wild-type Cdc12 (Supplementary Fig. 3d and Supplementary Table 1). Conversely, based on the data presented in Extended Data Fig. 8 of the original manuscript, we know that (as was originally stated in lines 149-154) "...given the N- and C-terminal regulatory regions are unchanged, mDia2-GFP properly co-localizes with Myo2's regulatory light chain Rlc1-tdTomato to rings [...and that formin] mDia2 supports cell division with modest cytokinesis defects...". With that in mind, we feel much more confident in testing the in vivo relevance of Cdc12's mechanosensitive (FH1-dependent) behavior by using the reported Cdc12N-mDia2FH1FH2-Cdc12C strain.

Instead, we therefore decided to focus our attention on a different experiment to directly test our hypothesis that myosin-mediated pulling inhibits Cdc12 through its FH1 domain. In the revised manuscript, we present new data from a set of multiple independent in vitro experiments (n=11 and n=8 for Cdc12 and mDia2, respectively) where instead of anchoring the formins Cdc12 and mDia2 via their FH1 domains (Figures 1-4), both formins were instead fixed to the coverslip surface via their FH2 domains (new Fig. 5). We found that Cdc12 remains active once myosin-mediated pulling strain is directly applied to Cdc12's FH2 domain, providing independent evidence that it is the FH1 domain that acts as Cdc12's force-sensing domain (new Fig. 5b). FH2 domain-attached mDia2 does not show a significant change in activity (new Fig. 5c) re-establishing that under the conditions tested mDia2 can be considered mechano-insensitive.

Furthermore, in the revised version we provide new direct evidence that the assembly of contractile rings of dividing *mDia2* hybrid cells is compromised with more than twice as much F-actin as determined for wild-type *cdc12* cells (see response to point 5b below).

- b. *The authors hypothesize that the hybrid protein causes more actin to build up at the nascent cytokinetic ring. Can they quantify actin at the ring to test the hypothesis, perhaps by rhodamine-phalloidin staining of fixed cells? Without this result, it is difficult to distinguish between their hypothesis (that mDia2 causes too much elongation) and simply having a somewhat poorly folded chimeric protein.*

Response: We fully agree with Rev.#2's concern that our hypothesis of *mDia2* cells generating an excess of F-actin at the equatorial plane compared to wild-type *cdc12* cells needed to be tested experimentally. We therefore quantified the amount of F-actin in the assembling ring of dividing *cdc12* and *mDia2* cells. As suggested by Rev.#2, we fixed and stained cells with BODIPY-Phalloidin and quantified the mean F-actin fluorescence of fully assembled rings that were not yet constricting (Fig. 7d,e; n=21 and n=20 for *cdc12* and *mDia2* cells; see p. 17 and 20 for details of the imaging and data analysis). In agreement with our hypothesis, contractile rings of dividing *mDia2* cells contain more than twice the amount of F-actin than wild-type *cdc12* cells (13.08 ± 0.73 vs. 6.03 ± 0.47 r.u., Median

Response to referees' comments

± S.E.M.), which supports our model that for dividing *mDia2* cells the observed "...node clumping likely arises from entanglement of nodes through excessive F-actin generation by [the mechano-insensitive and thus] uninhibited formin" (p.7, paragraph 1). Along with the mathematical modeling results of Fig. 7i of the revised manuscript, we therefore think that for the scope of this paper we put forth the required evidence arguing that FH1-dependent mechano-inhibition of Cdc12 facilitates proper ring assembly from precursor nodes in vivo.

6) Lines 173-175, discussing G-actin concentrations, come out of nowhere, and it is unclear what exactly their relevance is as written.

Response: In the original version, we thought it would be useful to remind the reader that due to the much higher cytoplasmic actin monomer concentration (compared to the 1.5 μ M used in our vitro experiments) formin-mediated filament elongation rates are known to be approx. 8-fold faster. Because it is technically not feasible to work at such high actin concentrations in vitro at single-filament level, the described mathematical model can provide important insight. However, we fully agree with Rev.#2, in that this point can be made without going into such detail while still adding to the clarity of the manuscript. We therefore amended this paragraph and included it into the discussion section of the revised manuscript (p.12 !?): "*Our mathematical modelling approach allowed us to further study the role of Cdc12's mechano-inhibition during contractile ring assembly under physiological conditions (i.e. 100-times higher drag forces and 8-times faster filament elongation rates) (Laporte et al. 2012, Luby-Phelps et al. 1987, Vavylonis et al. 2008).*"

7) *Figure 2B – can they have another example where there is a longer search phase?*

Response: We understand the reviewer's comment, but at the same time we prefer to use this example simply because the long capture period best demonstrates that active pulling by myosin is required to mechano-inhibit Cdc12. The strongly bound but not pulling myosin (for longer than 8 min, Fig. 2e,f) has no impact on Cdc12's ability to elongate actin filaments.

8) *Page 3. The elongation rates given in the text (10.6 and 11.4) are different than in Figure 1d and 2 b. Some clarification needed.*

Also, it would be best to give the mean elongation rates for at least the search filaments (and preferably all of the conditions) in the legends for Figure 1c, 2e and 2f (and the relevant graphs in figure 3).

In fact, why are the search filament points normalized individually for these plots? This makes it appear that there is no error at all in these readings. It would be more useful to have the spread of the elongation rates depicted for each of these points.

Response: Unless specified otherwise, all in-text rates have been updated with the median \pm S.E.M. values that are also summarized in the new Supplementary Table 1 of the revised manuscript.

All graphs do contain the rates (deduced from the regression fit line) for the respective example and all three (search, capture-pull, dissociation) phases. For all the relevant figure legends, we decided to include a reference to Supplementary Table 1, which contains all the median \pm S.E.M. values.

For the scope of this work we feel strongly that it is important to compare the formin's activity (i.e. filament elongation rate) during and after Capture-Pull directly to its baseline activity before the initial capture event. Due to the fact that between independent experiments, there is always some variability in the rates at which a given formin elongates filaments (see Supplementary Table 1 for variance), we think normalizing all events to their respective pre-capture formin activity is clearest. As mentioned above, for all the relevant figure legends, we have now included a reference to Supplementary Table 1, which contains all the median \pm S.E.M. values.

Response to referees' comments

We hope to have adequately addressed Reviewer#2's points of concern and would like to once again thank Reviewer #2 for her/his thoughtful comments and suggestions through which the quality of the revised manuscript is significantly improved.

Reviewers' comments:

Reviewer #1 (Remarks to the Author):

The authors have addressed most major issues I raised in my first review. A few remain.

1. Does visible filament length differ from actual length?

In my first review I noted abrupt jumps in Fig. 1d (filament length vs time) - a discontinuous decrease (increase) in apparent filament length immediately after capture (release). This suggests visible filament length during capture may not faithfully report the actin polymerization rate, and that the latter might in reality be unaltered by force or be altered by an amount differing from that suggested by the visible length during capture.

The point is, if you join up the blue and green curves with a straight line you get a much lower slope than the red line. It suggests hidden length that is recovered on release, and it suggests a much lower mechanosensitivity than implied by the red curve.

The issue is not dealt with in the revised manuscript (nor adequately in the rebuttal). However, Fig. 1d has been altered – the green curve has been pulled down so there is now almost no jump on release (see below). Otherwise, Fig. 1d is identical, i.e. the same data set is presumably being plotted. There is no mention of this alteration in the rebuttal or the manuscript.

Presumably the authors found out that the original Fig. 1d was in error, and the new version is now suitably corrected. If so, this needs to be clearly explained.

Furthermore, Fig. 1d is the only data presented that includes a release episode. What happens in other cases – is there always a jump on release, or never, or what?

Response to referees' comments

The authors have addressed most major issues I raised in my first review. A few remain.

1. Does visible filament length differ from actual length?

In my first review I noted abrupt jumps in Fig. 1d (filament length vs time) - a discontinuous decrease (increase) in apparent filament length immediately after capture (release). This suggests visible filament length during capture may not faithfully report the actin polymerization rate, and that the latter might in reality be unaltered by force or be altered by an amount differing from that suggested by the visible length during capture.

The point is, if you join up the blue and green curves with a straight line you get a much lower slope than the red line. It suggests hidden length that is recovered on release, and it suggests a much lower mechanosensitivity than implied by the red curve. The issue is not dealt with in the revised manuscript (nor adequately in the rebuttal).

However, Fig. 1d has been altered – the green curve has been pulled down so there is now almost no jump on release (see below). Otherwise, Fig. 1d is identical, i.e. the same data set is presumably being plotted. There is no mention of this alteration in the rebuttal or the manuscript.

Presumably the authors found out that the original Fig. 1d was in error, and the new version is now suitably corrected. If so, this needs to be clearly explained.

Furthermore, Fig. 1d is the only data presented that includes a release episode. What happens in other cases – is there always a jump on release, or never, or what?

Response:

We apparently did not completely understand the reviewer's primary concern (point #4) from the initial review, that the "jump" in the length of the filament shown in the original Fig. 1d might indicate that the elongation rate during capture-pull is not diminished. As described below, we had discovered that the "jump" was caused by a filament tracing error, which we apologize for not describing in our original rebuttal. We therefore thank the reviewer for bringing up this point again, which we agree warrants a thorough response.

The major point of the paper is that upon capture-pull by myosin we observe an instantaneous and significant (3 to 4-fold) decrease in the rate that the fission yeast cytokinesis formin Cdc12 processively elongates actin filaments. Hence, during capture-pull, actin subunits are still being added to the barbed filament end, albeit at a significantly slower rate. We determined the decrease in elongation rate by measuring filament length frame by frame. The filament length increases less quickly during capture-pull events, compared to before (Search) and after (Dissociation) Capture-Pull.

(1) In this manuscript, we report 33 events in which the Cdc12-mediated filament elongation rate decreases upon capture-pull, whether Cdc12 is adhered to a bead (Fig. 1, n=13) or adhered directly to the coverslip surface (Fig. 2, n=20). The decrease in elongation rate exactly coincides with co-localization of the Myosin-coated bead to the filament and subsequent bead/filament translocation. There are only three (of 20) events in which the coverslip-bound Cdc12-mediated filament elongation rate does not decrease upon capture-pull (top three red squares in Fig. 2d), and none in the 13 cases of formin bound to a bead (Fig. 1e).

Furthermore, we included numerous controls in which bead capture does not decrease the formin-mediated elongation rate, strongly indicating that the inhibition we see with Cdc12 is not artefactual. These controls include inactive (NEM-treated) myosin (Fig. 2e-g), mammalian formin mDia2 (Fig. 3a-c), chimera formin mDia2(FH1)-Cdc12(FH2) (Fig. 4b) and new data showing that Cdc12 adhered to the surface via its FH2 domain rather than its FH1 domain (Fig. 5a-b).

(2) The reviewer's reasonable skepticism had arisen from the fact that in the original version, the particular example that we chose for Fig. 1d had a "jump" between the Capture-Pull phase and the dissociation phase. We agree with the reviewer that it would be theoretically possible that the "jump" indicates that the elongation rate during the capture-pull phase was not significantly decreased.

Response to referees' comments

Unfortunately, as this turned out to be a measurement error (as will be explained below), we focused our attention on the reviewer's many other points and did not do an adequate job of addressing this concern in our initial rebuttal. We had chosen the example shown in Fig. 1b-d because it is a representative example showing that (1) inhibition coincides with the onset of Capture-Pull, (2) the rates of formin Cdc12-mediated filament elongation before (Search) and after (Dissociation) Capture-Pull are similar, indicating that the inhibitory effect is specific to Capture-Pull and reversible once the myosin dissociates from the filament, (3) myosin-mediated pulling drives the coalescence of the formin- and myosin-associated beads, reminiscent of two neighboring nodes coalescing in vivo, and (4) the movie is relatively clear of other filaments allowing nice visualization of the various events.

Upon reviewer #1 pointing out that the initially submitted Fig. 1d had a "jump" in filament length between Search-Capture and Dissociation, we looked carefully and confirmed that there were NO "jumps" between the Search-Capture and Dissociation of any of the other 22 Cdc12-mediated assembly events in which the myosin bead dissociated from the filament. Therefore, for the original revision we looked into the cause of the "jump" in the original version of Fig. 1d. We discovered that the "jump" was an error that occurred during filament tracing, caused by an incorrectly set pointed end of the filament (due to a heavy load of filaments in the region), which added extra filament length to measurements following Dissociation. When the correct filament pointed end was used, the "jump" disappeared, but importantly the relative filament elongation rates did not change. We apologize for not specifically pointing this out in the initial rebuttal letter, which was simply an oversight as we were focused on the reviewer's other concerns/questions/suggestions. Again, we thank the reviewer for pointing out the "jump" in the original version of Fig. 1d, and apologize for not having explained the corrected version of Fig. 1d in the revised manuscript.

(3) Over more than a decade the Kovar lab has published numerous papers utilizing single filament TIRF microscopy to characterize the processive elongation behavior of numerous formins. Tracing filaments and reporting formin-mediated F-actin elongation rates is a key expertise of ours. In all these years, we have never come across such a statistically significant inhibition in formin elongation rate. If our methods for measuring filament length over time are inherently error-prone, we would expect to have encountered such "inhibition artifacts" before.

(4) We are confident that we have accurately measured and reported actin elongation correctly throughout the manuscript and apologize once again for not pointing out the correction of Fig. 1d in our initial rebuttal letter and for any confusion this may have caused the referees to interpret our findings adequately.

2. Comparing Cdc12 mechanosensitivity to other formins

I remarked that the authors results (first submission) implied a formin Cdc12 mechanosensitivity ~700 times that reported for mDia1 by Jegou et al, 2013 (and of opposite sign). I suggest this factor (or whatever is the correct factor) be stated in the manuscript, because such a huge difference is surprising.

In the revised manuscript, the authors report a ~4 fold lower polymerization rate for an estimated 0.1 pN force and write: "...under hydrodynamic flow, tensile forces in the range of 0.1–2.5 pN can accelerate formin Bni1- (budding yeast) and mDia1-mediated (mammalian) F-actin elongation rates ~2-fold", referencing Courtemanche et al., 2013 and Jegou et al., 2013.

However, the quantitative comparison is not made. Compared to Bni1 and mDia1, how many fold larger is Cdc12 mechanosensitivity according to this study?

I estimate that in both of these prior studies the mechanosensitivity was roughly 0.5 fold per pN, (Fig. 4C and Fig. 3C in Courtemanche and Jegou, respectively). Thus, the presently claimed mechanosensitivity is ~ 80 times higher (not 13 times as the authors claim in the rebuttal).

Response to referees' comments

Response:

We appreciate the referee's comment that we should not simply re-state the absolute fold-change in activity of formins Bni1 and mDia1 over the force range those formins were tested, but rather compare the fold-changes in formin activity per pN for Cdc12 to those of Bni1 and mDia1. We agree that the fold-change pN^{-1} parameter would be a nice way to compare the mechanosensitivities of the different formins. However, due to the fact that the force response for formins mDia1 and Bni1 is non-linear (personal communication with A. Jegou, first-author of the mDia1 Nat. Comm. 2013 paper) with the elongation rate doubling and saturating at ~ 3 pN, we think one should be cautious in how such a quantity is reported. To incorporate the referee's good suggestion, we have added the following paragraph to provide the reader with a more direct comparison between formins mDia1/Bni1 and Cdc12 (bottom of p.8):

For formins mDia1 and Bni1, the response to flow-induced mechanosensitivity occurs in a non-linear fashion (referencing Courtemanche et al., 2013 and Jegou et al., 2013), making it difficult to directly compare the mechanosensitivities of those formins to that of Cdc12. However, it should be mentioned that formin Cdc12 responds to force in a manner that is not only of the opposite sign but also seems to be much more sensitive to force than formins Bni1 and mDia1 (3- to 4-fold change at ~ 0.1 – 1 pN vs. ~ 2 -fold change in activity over a 3 pN range) (referencing Courtemanche et al., 2013 and Jegou et al., 2013). This force sensitivity, which could be >60 -times higher per pN is perhaps reflective of the difference in mechanism between the FH1-mediated mechanosensitivity of formin Cdc12 and the presumed FH2-dependent mechanosensitivity of formins Bni1 and mDia1 (referencing Courtemanche et al., 2013 and Jegou et al., 2013).

3. Estimating bead drag coefficients to get the force acting on the formin

In Methods, all bead trajectories are fitted to a diffusive dynamics, i.e. with linear dependence of mean square displacement on time. The Einstein relation is used to extract the bead drag coefficient for each trajectory. Since about 75% of trajectories are sub-diffusive, this is an invalid procedure. I agree that this is a reasonable, practical measure to take, given noisy data. However, strictly this is incorrect and this should be stated in the manuscript.

Response:

We are in perfect agreement with the referee. We also feel this is a "reasonable, practical measure to take" in order to have an estimate of the drag forces on beads, while fully realizing that the situation is more complex within the experiment. We have changed the sentence in the discussion to strongly emphasize that we are making this large approximation (as addressed in the methods section).

The discussion (p.9, paragraph 1) now reads:

However, in 75% of cases the beads showed sub-diffusive behaviour, which we attribute to non-specific interactions of the beads with the coverslip. Given that these data are noisy and we do not have strong evidence for the precise mechanism underlying this sub-diffusion, we fit data from these sub-diffusing beads as if they are diffusing in order to extract an approximate average viscosity/drag coefficient experienced by the beads, although this procedure is not strictly correct. The effective viscosity extracted in this way is $\eta=0.22$ Pa s, which is ~ 20 -fold higher than our estimate of the viscosity of the medium.